

# Spectral statistics of a minimal quantum glass model

Richard Barney[1], Michael Winer[1], Christopher L. Baldwin[1],
Brian Swingle[2] and Victor Galitski[1,3]

**1** Joint Quantum Institute, Department of Physics, University of Maryland,
College Park, Maryland 20742, USA
**2** Department of Physics, Brandeis University, Waltham, Massachusetts 02453, USA
**3** Center for Computational Quantum Physics, The Flatiron Institute,
New York, NY 10010, USA

## Abstract

Glasses have the interesting feature of being neither integrable nor fully chaotic. They thermalize quickly within a subspace but thermalize much more slowly across the full space due to high free energy barriers which partition the configuration space into sectors. Past works have examined the Rosenzweig-Porter (RP) model as a minimal quantum model which transitions from localized to chaotic behavior. In this work we generalize the RP model in such a way that it becomes a minimal model which transitions from glassy to chaotic behavior, which we term the "Block Rosenzweig-Porter" (BRP) model. We calculate the spectral form factors of both models at all timescales larger than the inverse spectral width. Whereas the RP model exhibits a crossover from localized to ergodic behavior at the Thouless timescale, the new BRP model instead crosses over from glassy to fully chaotic behavior, as seen by a change in the steepness of the ramp of the spectral form factor.



# 1   Introduction

Random Hermitian matrices provide a simple model for the energy levels of a wide variety of quantum systems, including complex nuclei [1–6], systems with a chaotic classical limit [7–12], strongly interacting quantum field theories [13–17], and much more. The wide prevalence of random-matrix-like energy levels is known as random matrix universality [7, 18, 19], and it is one of the key manifestations of quantum chaos. Given an ensemble of Hamiltonians $\{H\}$, one way to characterize random matrix universality is in terms of the statistical correlations of eigenvalues. Letting $\rho(E)$ denote the density of eigenvalues of a particular random $H$, then although the average density $\overline{\rho(E)}$ is not universal, the pair-correlation $\overline{\rho(E)\rho(E')}$ does turn out to be. Indeed, one finds that the pair-correlation is closely related to the pair-correlation of a Gaussian random matrix of the appropriate symmetry.

However, while many systems ultimately exhibit random-matrix-like behavior at the finest energy scales, i.e. for sufficiently small $E - E'$, real systems typically have additional structure in their energy spectrum which is not random-matrix-like. This structure may be eventually washed out at the finest energy scales, but it does cause a deviation from the random matrix behavior of $\overline{\rho(E)\rho(E')}$ when $E - E'$ is of some intermediate size. Perhaps the simplest such structure occurs when the Hamiltonian breaks up into approximately decoupled blocks labelled by some almost-conserved quantity. This situation is common and is broadly related to the presence of slow dynamics, e.g. a slowly diffusing charge or almost-frozen glassy dynamics. In the simplest case, each block is statistically independent and the blocks are connected by some additional weak perturbations. Then as a function of $E - E'$, the pair-correlation can exhibit a crossover from multiple small random blocks to a single large random block. Recently, a set of effective theories have been formulated to describe this crossover [20]. Here we present a new solvable model in which this crossover can be analytically verified and studied. The results are consistent with Ref. [20], but they also offer new insights into various regimes.

This new model is a generalization of the unitary Rosenzweig-Porter (RP) model, which is itself a slight generalization of the original model constructed by Rosenzweig and Porter to describe complex atomic spectra [21]. The RP Hamiltonian is

$$H = A + \frac{\lambda}{N^{\gamma/2}} V, \tag{1}$$

where $A$ is a diagonal matrix with independent and identically distributed elements, and $V$ is a random matrix drawn from the Gaussian Unitary Ensemble (GUE) [22] whose matrix elements have unit variance. $N$ is the size of the Hilbert space. Several studies [23–29, 29–37] have examined this model at different values of $\gamma$. The RP model has also been generalized in several different ways to create a family of interesting random matrix ensembles [38–44].

Motivated by the problem of many-body localization, Kravtsov et al. [27] studied this RP ensemble as a simple random matrix model with both localization and ergodicity-breaking

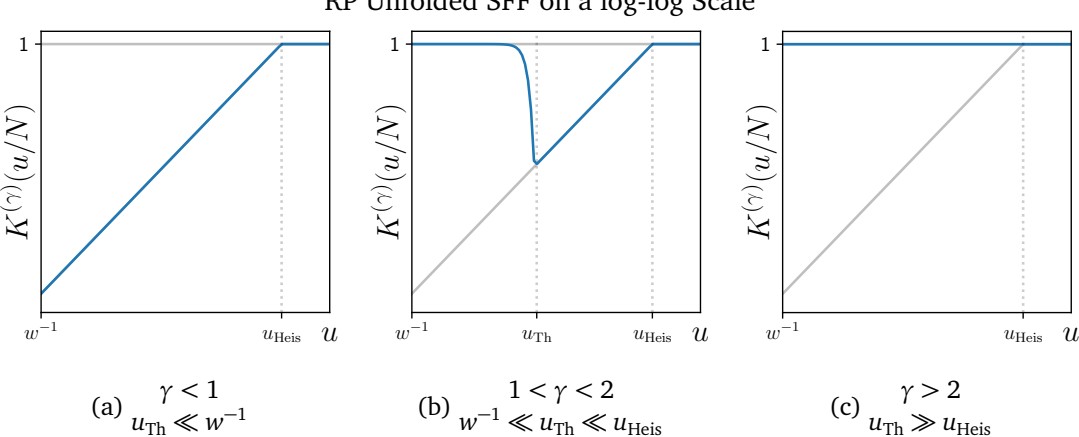

Figure 1: Schematic representation on a log-log scale of the unfolded SFF for the RP model in each of the three identified phases. The lower gray curve is the GUE SFF while the upper gray curve is the Poissonian result.

transitions. Consistent with previous results [23–26, 45], they found Poissonian statistics for $\gamma > 2$, indicating Anderson localized behavior, and GUE statistics for $\gamma < 1$, indicating chaotic behavior. They found that intermediate values of $\gamma$ led to a non-ergodic extended phase in which eigenstates are neither localized nor fully ergodic.

One useful tool for studying spectral statistics is the unfolded spectral form factor (SFF), which is the Fourier transform of the unfolded two-point correlation function. Unfolding refers to the process of rescaling the spectrum in such a way that the local mean level spacing becomes unity, bringing the universal features of the SFF to the fore. Ref. [27] provides a calculation of the RP SFF for all $\gamma > 1$ at the Thouless timescale, i.e., the timescale at which random matrix statistics first appear. Other timescales are then examined by rescaling time in this result.

We can form schematic expectations for the behavior of the RP SFF at different values of $\gamma$ by comparing the Thouless time to the two other cardinal times: the inverse spectral width $w^{-1}$ and the Heisenberg time, which is the inverse mean level spacing. These schematics are shown in Fig. 1. In the figure $u/N$ is the unfolded time, that is the real time divided by the level density. When the Thouless time is smaller than the inverse spectral width, which occurs when $\gamma < 1$, we expect GUE statistics indicating chaotic behavior. When it is larger than the Heisenberg time, which occurs when $\gamma > 2$, we expect Poissonian statistics indicating localized behavior. If the Thouless time is larger than the inverse spectral width but smaller than the Heisenberg time we expect to see a crossover between the two behaviors. Phase transitions occur when the Thouless timescale coincides with one of the cardinal timescales.

As an initial result of the present work, we directly calculate the SFF of the RP model at all timescales larger than the inverse spectral width. This approach has the added benefit of more closely paralleling the calculation of the SFF for the new model we examine. The result is shown in Eq. (66). We find that this result does follow the schematic expectations shown in Fig. 1 and is in good agreement with the numerical results we obtain. All these findings are in agreement with Ref. [27].

In the bulk of this work, we generalize the RP model to obtain a random matrix model which transitions from chaotic to glassy behavior, where glassiness is identified by the presence of multiple thermalization timescales. This is accomplished by redefining $A$ in Eq. (1) to be the block-diagonal matrix

$$A = \bigoplus_{i=1}^{P} A^{(i)}, \tag{2}$$

where each block $A^{(i)}$ is an independent GUE matrix of size $M = N/P$ whose elements have variance $M^{-1}$ (so that the eigenvalues of each $A^{(i)}$ are finite at large $M$). We call this generalization of the RP model the Block Rosenzweig-Porter (BRP) model. Because of the the normalization of each $A^{(i)}$ the system thermalizes in $O(1)$ time within each block but can fully thermalize only at times diverging with $N$ (if at all). Due to the different structure of the $A$ matrix, the BRP model will have a localized phase different from that of the RP model. In this phase the system will be confined to the subspaces corresponding to the blocks of $A$ instead of individual states.

This generalization of the RP model is motivated by our recent work examining the SFF of a canonical quantum spin glass model [46]. In that work we found that the SFF has a linear ramp at times sub-exponential in the system size, as would a chaotic system. However, the ramp is steeper by a factor of the number of distinct metastable states. This enhancement of the ramp is a hallmark of glassy behavior. It indicates that, at these early timescales, the Hamiltonian can be viewed as a direct sum of independent random matrices. However, it has been argued that, for appropriate values of the coupling between spins and at a sufficiently late timescale, the system will escape from the metastable configurations and fully thermalize [47–49]. At this timescale we would expect the SFF to experience a crossover from the enhanced ramp to the GUE ramp. Due to the challenge of calculating the SFF of canonical quantum spin glass models at late times with current methods, we turn to the BRP model as a simpler model of a quantum glass for which the SFF can be calculated at late timescales and the crossover from glassy to chaotic behavior can be seen.

As we did for the RP model, we form schematic expectations for the behavior of the SFF at different values of $\gamma$ by comparing the Thouless time with the other cardinal times. These schematics are shown in Fig. 2. When the Thouless time is less than the inverse spectral width, which we find occurs when $\gamma < 1$, we expect GUE statistics indicating chaotic behavior. When it is larger than the Heisenberg time, which we find occurs when $\gamma > 2$, we expect the statistics of uncoupled GUE blocks, indicating localization within the blocks. Note that in this case the SFF reaches its plateau value at a time earlier than the Heisenberg time. This is the block Heisenberg time, which is the inverse mean level spacing for a single block of $A$. This means that the block Heisenberg time is smaller than the full Heisenberg time by a factor of $P$, the number of blocks. The block Heisenberg time is an additional cardinal time in the BRP model.

If the Thouless time is larger than the inverse spectral width and smaller than the Heisenberg time we expect to see a crossover between the block localized and chaotic behaviors at the Thouless timescale. Further, if the Thouless time is smaller than the block Heisenberg time, which occurs when $\gamma < 1 + d$ where $d = \log_N M$, this crossover will occur before the SFF can reach its plateau value at the block Heisenberg time. If the Thouless time is greater than the block Heisenberg time, which occurs when $\gamma > 1 + d$, the SFF will reach its plateau value at the block Heisenberg time, drop down to the GUE result at the Thouless timecale, then reach the plateau value again at the Heisenberg time.

Our primary result is the calculation of the SFF of the BRP model at all timescales larger than the inverse spectral width, which is $O(1)$. This result is shown in Eqs. (132)-(133). It follows the schematic expectations shown in Fig. 2 and is in good agreement with the numerical results we obtain. At the Thouless timescale, if it is not of the same order as the Heisenberg time, the SFF decays exponentially over time from its glassy result to the GUE SFF, as shown in Eq. (155). In the case where the Thouless time is of the same order as the Heisenberg time, a more complicated crossover behavior emerges, shown in Eqs. (156)-(157).

Our results indicate that for $\gamma < 1$ the system will immediately thermalize while for $\gamma > 2$ the system will always remain localized within blocks. For intermediate values of $\gamma$ the system will initially be confined to a single block, but will escape and thermalize at the Thouless time. We also find a transition at $\gamma = 1 + d$. This transition has an important interpretation in terms

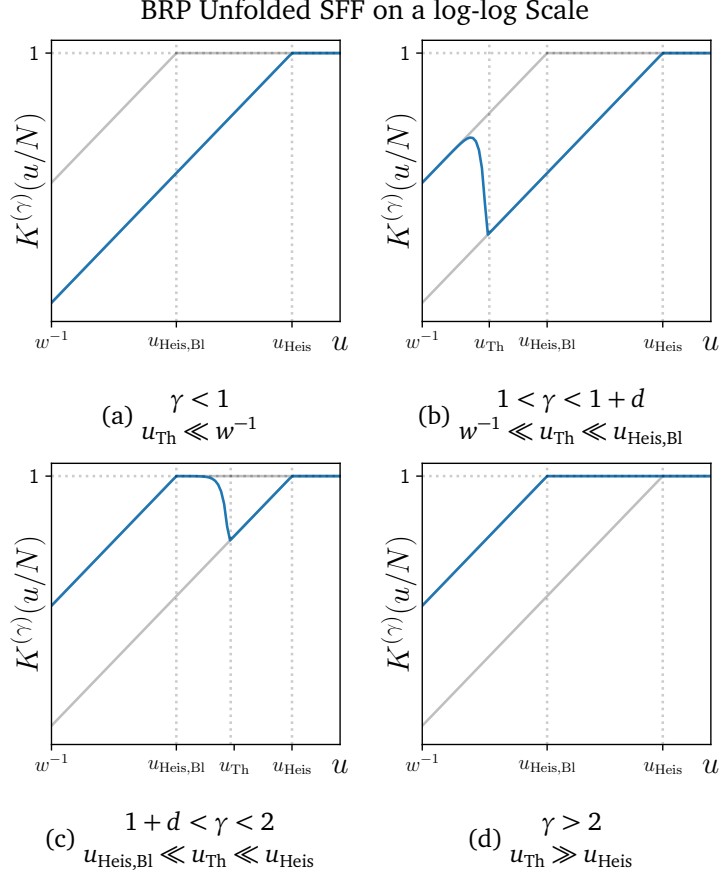

Figure 2: Schematic representation on a log-log scale of the unfolded SFF for the BRP model in each of the four identified phases. The lower gray curve is the GUE SFF while the gray curve above it is the SFF for completely uncoupled GUE blocks.

of the eigenstates. It is the point at which the eigenstates become fully delocalized across the Hilbert space.

The remainder of this work is organized as follows. In section 2 we review the spectral form factor's definition, features, and role as a diagnostic of quantum chaos. In section 3 we outline the aspects of the construction of the SFF which are common to both the RP and BRP models and discuss the unfolding procedure. In section 4 we examine the RP SFF, discussing the results of previous works and directly calculating the SFF at relevant timescales. We compare our analytical results with numerics. Section 5 contains the bulk of our results, an examination of the BRP model and the calculation of its SFF, again comparing to numerical results. Finally, we conclude with section 6.

## 2 Review of the spectral form factor

The spectral form factor (SFF) has a long history as a diagnostic of quantum chaos [7,18,22]. Examples of random-matrix-like SFFs in chaotic systems appear in numerous areas of physics, from nuclear systems [6,50] to condensed matter [51–53] to holographic theories [11,12]. The SFF diagnoses whether energy levels repel as they do in random matrices [51], have independent Poissonian statistics [54], or have some more exotic behavior [55–57]. The SFF can be written as

$$\text{SFF}(t, f) = \overline{|\text{tr}[e^{-iHt} f(H)]|^2}, \tag{3}$$

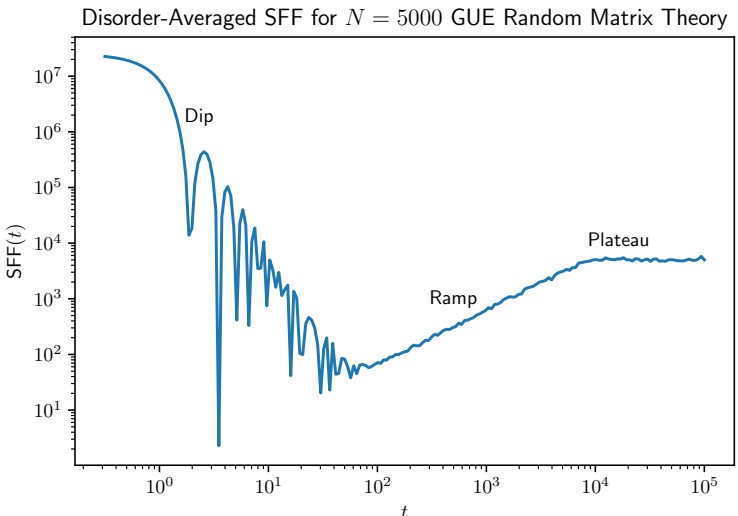

Figure 3: A log-log plot of the disorder-averaged SFF for the Gaussian unitary ensemble (GUE). The matrices in this ensemble have dimension $N = 5000$. The SFF was computed numerically by exactly diagonalizing five hundred realizations. The three regimes of the SFF—dip, ramp, plateau—are each labeled.

where $f$ is a filter function used to pick an energy band of interest, and the overline denotes a disorder average over an entire ensemble of Hamiltonians.

The SFF can also be expressed as the two-point function of the density of states. Defining

$$\rho(E, f) = \sum_n \delta(E - E_n) f(E_n) = \text{tr}\left(\delta(H - E) f(H)\right), \tag{4}$$

we have

$$\overline{\text{SFF}(t, f) = \int dE_1 dE_2 \rho(E_1, f) \rho(E_2, f) e^{i(E_1 - E_2)t}}. \tag{5}$$

As a two-point function, the SFF can be broken down into connected and disconnected components. These components have very different behaviors, as discussed below.

For random-matrix-like systems, the spectral form factor has three regimes of interest.

- The "dip", also known as the slope, occurs at early times. It comes from the disconnected piece of the SFF (and thus its precise shape is non-universal and depends on the details of $f$ and the thermodynamics of the system). Its downward nature reflects a loss of constructive interference: at $t = 0$ the terms in $\text{tr}[e^{-iHt} f(H)]$ are all positive, but the different terms of $\text{tr}\, e^{-iHt}$ acquire different phase factors as $t$ increases.

- The "ramp" occurs at intermediate times. It is arguably the most interesting regime, and marks the beginning of the universal behavior in the connected spectral form factor. In the canonical matrix ensembles, it is a consequence of the result [22]

$$\mathbb{E}\left[\rho\left(E + \frac{\omega}{2}\right)\rho\left(E - \frac{\omega}{2}\right)\right] - \mathbb{E}\left[\rho\left(E + \frac{\omega}{2}\right)\right]\mathbb{E}\left[\rho\left(E - \frac{\omega}{2}\right)\right] \sim -\frac{1}{\mathfrak{b}\pi^2\omega^2}, \tag{6}$$

where $\mathfrak{b} = 1, 2, 4$ for the orthogonal, unitary, and symplectic ensembles respectively. The fact that the right hand side is negative is a manifestation of level repulsion [50]. Taking the Fourier transform of Eq. (6) with respect to $\omega$ gives a term proportional to $t$

for the connected SFF. Such a linear-in-$t$ ramp is often taken as a defining signature of quantum chaos.

The exact coefficient of the ramp can tell us a lot about a quantum system. For instance, if $H$ is not a GUE matrix but the direct sum $H_1 \oplus H_2$ of two GUE matrices, the ramp will be enhanced by a factor of two. This enhancement shows up in realistic systems such as the Bunimovich stadium [58, 59], which have different sectors of their Hamiltonian which behave differently under reflection symmetry. It has recently been shown [46] that at times sub-exponential in system size, all-to-all spin glasses exhibit an enhancement of the ramp equal to the number of effective "sectors", i.e., regions of configuration space rendered dynamically disconnected by large energy barriers. This work serves as an extension of that result for a toy model of spin glasses, going out to late times.

- The "plateau" occurs at late times. It is a signature of the discreteness of the spectrum. It is part of the connected spectral form factor and is completely universal to all systems, whether thermalizing, integrable, glassy, or many-body localized [60, 61]. At times much larger than the inverse level spacing or "Heisenberg time", one expects that all off-diagonal terms in the double-trace of the SFF average to zero, meaning that

$$\text{SFF}(t, f) = \sum_{mn} \overline{e^{-i(E_m - E_n)t} f(E_m) f(E_n)} \sim \sum_n f(E_n)^2 \,. \tag{7}$$

For integrable systems, the plateau is reached very quickly with little to no ramp regime [54, 56, 57], whereas for chaotic systems the plateau isn't reached until a time exponential in system size.

In the bulk of this paper we calculate the unfolded connected SFF for the RP and BRP models at all timescales larger than the inverse spectral width, thus we capture both the ramp and the plateau regimes. We find that when the sectors of the BRP model are uncoupled the ramp of the SFF is enhanced by a factor of the number of sectors. As the coupling strength is increased there will be a crossover to the regular ramp at the Thouless timescale.

## 3 Construction of the spectral form factor

In this section we lay out the initial steps of the construction of the SFF which are common to both the RP model and the new BRP model. In fact, the results of this section are applicable to any Hamiltonian which is perturbed by a GUE matrix. Both of the models can be expressed as a Hamiltonian matrix of size $N$ with the form

$$H = A + V \,, \tag{8}$$

where the spectrum of $A$ is centered at 0 and has a width that is $O(1)$ with respect to $N$. $V$ is a random GUE matrix drawn from the distribution

$$p_V(V) \sim \exp\left(-\frac{1}{2\sigma} \operatorname{tr} V^2\right), \qquad \sigma = \frac{\lambda^2}{N^\gamma} \,. \tag{9}$$

The exact details of $A$ will depend on whether we are working with the RP or BRP ensemble, but are unimportant at this initial stage.

### 3.1 The joint probability density function

Our first step is to find the joint probability density function (JPDF) of the eigenvalues of the Hamiltonian $H$. We follow the method of Kunz and Shapiro [45]. For the moment we will hold $A$ constant. The probability distribution of $H$ is then

$$p_H(H) = p_V(H - A) \sim \exp\left[-\frac{1}{2\sigma}\sum_i \left(E_i^2 + a_i^2\right)\right]\exp\left(\frac{1}{\sigma}\operatorname{tr} AUEU^\dagger\right), \tag{10}$$

where $\{a_i\}$ are the eigenvalues of $A$, $\{E_i\}$ are the eigenvalues of $H$, $E = \operatorname{diag}(E_1,\ldots,E_N)$ is the diagonal matrix similar to $H$, and $U$ is the unitary matrix which diagonalizes $H$.

We now make the change of variables $H \to \{U, E\}$. The Jacobian of this change is $\Delta^2(E)$, where $\Delta(E) = \prod_{i>j}(E_i - E_j)$ is the Vandermonde determinant. The JPDF is

$$p(E) \sim \Delta^2(E)\exp\left[-\frac{1}{2\sigma}\sum_i\left(E_i^2 + a_i^2\right)\right]\int dU \exp\left(\frac{1}{\sigma}\operatorname{tr} AUEU^\dagger\right), \tag{11}$$

where $dU$ is the Haar measure over the unitary group $U(N)$. We can evaluate the integral over $U$ using the Itzykson-Zuber integral identity [14]

$$\int dU \exp\left(\frac{1}{\sigma}\operatorname{tr} AUEU^\dagger\right) \sim \frac{\det \exp(a_i E_j/\sigma)}{\Delta(a)\Delta(E)}. \tag{12}$$

Applying this identity, we find that the JPDF is

$$p(E) = c(\sigma, N)\frac{\Delta(E)}{\Delta(a)}\exp\left[-\frac{1}{2\sigma}\sum_i\left(E_i^2 + a_i^2\right)\right]\det\exp\left(\frac{a_i E_j}{\sigma}\right), \tag{13}$$

with $c(\sigma, N)$ to be determined by normalization.

To normalize we calculate

$$\begin{aligned}
1 = \int dE\, p(E) &= \frac{c(\sigma, N)}{\Delta(a)}\int dE\,\Delta(E)\exp\left[-\frac{1}{2\sigma}\sum_i\left(E_i^2 + a_i^2\right)\right]\det\exp\left(\frac{a_i E_j}{\sigma}\right)\\
&= \frac{c(\sigma, N)}{\Delta(a)}\int dE\,\Delta(E)\exp\left[-\frac{1}{2\sigma}\sum_i\left(E_i^2 + a_i^2\right)\right]\sum_{\pi\in S_N}\operatorname{sign}(\pi)\prod_{i=1}^N\exp\left(\frac{a_i E_{\pi(i)}}{\sigma}\right)\\
&= c(\sigma, N)\frac{N!}{\Delta(a)}\int dE\,\Delta(E)\exp\left[-\frac{1}{2\sigma}\operatorname{tr}(E - a)^2\right],
\end{aligned} \tag{14}$$

where $a = \operatorname{diag}(a_1, \ldots, a_N)$. In the second line above $S_N$ is the set of permutations of $N$ elements. The third line follows because $\Delta(E)$ is antisymmetric under all transpositions. We now use the identity (proven in the appendix of [45])

$$\int dE\,\Delta(E)\exp\left[-\frac{1}{2\sigma}\operatorname{tr}(E - a)^2\right] = \Delta(a)(2\pi\sigma)^{N/2}, \tag{15}$$

to find that the normalization factor is

$$c(\sigma, N) = \frac{1}{N!}(2\pi\sigma)^{-N/2}. \tag{16}$$

We now let $A$ also be a random matrix. From Eqs. (13) and (16) we see that, for any function $W(E)$ which is symmetric in the eigenvalues of $H$, the ensemble average is

$$\overline{W}(E) = (2\pi\sigma)^{-N/2}\left\langle\frac{1}{\Delta(a)}\int dE\,W(E)\Delta(E)\exp\left[-\frac{1}{2\sigma}\operatorname{tr}(E - a)^2\right]\right\rangle, \tag{17}$$

where the angle brackets indicate that the average over the distribution of $A$ still needs to be performed. Fortunately the SFF and the correlation functions examined below are such symmetric functions.

## 3.2 Correlation functions

We now examine two correlation functions which we will use to construct the SFF. The first is the Fourier transform of the level density

$$\overline{C_1}(t) = \sum_k \overline{e^{itE_k}}. \tag{18}$$

Averaging with respect to the JPDF and making use of Eq. (15) yields

$$\overline{C_1}(t) = e^{-\sigma t^2/2} \left\langle \sum_k e^{ita_k} \frac{\Delta(a+\delta_k\tau)}{\Delta(a)} \right\rangle = e^{-\sigma t^2/2} \left\langle \sum_k e^{ita_k} \prod_{j \neq k} \left(1 + \frac{\tau}{a_k - a_j}\right) \right\rangle, \tag{19}$$

where $\tau = it\sigma$ and $\delta_k$ is the projection matrix onto the $k^{\text{th}}$ dimension. In order to average over the eigenvalues of $A$ we rewrite this as

$$\overline{C_1}(t) = \frac{e^{-\sigma t^2/2}}{\tau} \oint_{\mathcal{C}} \frac{dz}{2\pi i} e^{itz} \left\langle \prod_j \left(1 + \frac{\tau}{z - a_j}\right) \right\rangle, \tag{20}$$

where $\mathcal{C}$ is a rectangular integration contour of infinitesimal width in the imaginary direction which encompasses the real axis. Using this form is advantageous because each term within the angle brackets now depends on only a single eigenvalue of $A$.

We now consider the Fourier transform of the two-point correlation function (excepting the $k = l$ terms)

$$\overline{C_2}(t) = \sum_{k \neq l} \overline{e^{it(E_k - E_l)}}. \tag{21}$$

Again, averaging with respect to the JPDF and using Eq. (15) yields

$$\begin{aligned}
\overline{C_2}(t) &= e^{-\sigma t^2} \left\langle \sum_{k \neq l} e^{it(a_k - a_l)} \frac{\Delta(a + \tau(\delta_k - \delta_l))}{\Delta(a)} \right\rangle \\
&= e^{-\sigma t^2} \left\langle \sum_{k \neq l} e^{it(a_k - a_l)} \left[1 - \left(\frac{\tau}{a_l - a_k - \tau}\right)^2\right] \prod_{j \neq k} \left(1 + \frac{\tau}{a_k - a_j}\right) \prod_{j \neq l} \left(1 - \frac{\tau}{a_l - a_j}\right) \right\rangle.
\end{aligned} \tag{22}$$

We can also write this in terms of contour integrals as

$$\overline{C_2}(t) = -\frac{e^{-\sigma t^2}}{\tau^2} \oint_{\mathcal{C}} \frac{dz}{2\pi i} \oint_{\mathcal{C}} \frac{dz'}{2\pi i} e^{it(z-z')} \left[1 - \left(\frac{\tau}{z'-z-\tau}\right)^2\right] \left\langle \prod_j \left(1 + \frac{\tau}{z-a_j}\right)\left(1 - \frac{\tau}{z'-a_j}\right) \right\rangle. \tag{23}$$

## 3.3 The unfolded spectral form factor

We now combine the correlation functions considered above to form the function

$$C(t) = \frac{1}{N}\left(\overline{C_2}(t) - \left|\overline{C_1}(t)\right|^2\right), \tag{24}$$

which is the connected SFF apart from the missing $k = l$ terms in $C_2(t)$. We will now demonstrate how $C(t)$ can be used to find the unfolded SFF. We note that

$$C(t) = -\frac{1}{N}\int dz\, dz'\, e^{it(z-z')}\left(\rho(z)\rho(z') - \rho_2(z,z') + \sum_k \overline{\delta(z - E_k)\delta(z' - E_k)}\right), \tag{25}$$

$$\rho(z) = \sum_k \overline{\delta(z - E_k)}, \quad \rho_2(z,z') = \sum_{k,l} \overline{\delta(z - E_k)\delta(z - E_l)}. \tag{26}$$

We now unfold by making the change of variables

$$(z, z') = x \pm \frac{y}{2\rho(x)}, \qquad (27)$$

$$t = \rho(x)T. \qquad (28)$$

In these new variables $x$ is the central energy and $y$ is the distance between the energies in units of local mean level spacings at the central energy. With this change we find

$$C(t) = \int dx \, p(x)(K(x, T) - 1), \qquad (29)$$

$$K(x, T) = -\int dy \, Y(x, y)e^{iTy}, \qquad (30)$$

$$Y(x, y) = \frac{1}{[\rho(x)]^2}\left[\rho\left(x + \frac{y}{2\rho(x)}\right)\rho\left(x - \frac{y}{2\rho(x)}\right) - \rho_2\left(x + \frac{y}{2\rho(x)}, x - \frac{y}{2\rho(x)}\right)\right], \quad (31)$$

where $p(x) = \rho(x)/N$ is the probability distribution function, $Y(x, y)$ is the unfolded connected two-point correlation function, also called the unfolded two-point cluster function, and $K(x, T)$ is the unfolded connected SFF. Our approach, whichever model we are examining, will be to put $C(t)$ in the form of Eq. (29) and extract the unfolded connected SFF, which we will simply call the SFF going forward.

For the models we study in this work it is reasonable to assume that $Y(x, y)$ vanishes in the large $N$ limit for $y \gg 1$, so we can restrict the interval of integration in Eq. (30) to be of order 1. For these values of $y$ the disconnected part of $Y(x, y)$ will tend to 1, thus it will not contribute to the SFF for $T \neq 0$. In the calculations that follow we will therefore neglect the disconnected piece of the two-level cluster function. This is an advantage of the unfolding procedure; we no longer need to worry about subtracting out the disconnected part in order to observe the universal features of the SFF. We will also find that if we unfold the coupling parameter $\lambda$ as well, $Y(x, y)$ and the SFF will become independent of the center energy $x$. So, although $C(t)$ is not universal due to its dependence on $p(x)$, the unfolded connected two-point correlation function and the SFF will be universal.

It will be helpful to use the time variable $u = N|T| = |t|/p(x)$, so $u$ is of the same order as the real time $t$. This means that $u$ is the time unfolded by the level probability density instead of the level density. The modulus may be taken since the SFF is symmetric in time. Because the SFF is the Fourier transform of the two-point cluster function, at time $u$ it probes the correlations between energy levels with separations on the order of $u^{-1}$. For this reason we consider only timescales larger than the inverse spectral width.

## 4 The Rosenzweig-Porter model

The Hamiltonian matrix for the RP model has the form of Eq. (8), with the additional condition that the eigenvalues of $A$ are all independent and identically distributed. We can consider $A$ to be diagonal such that

$$A = \text{diag}(a_1, \ldots, a_N), \qquad (32)$$

where each $a_i$ is independently drawn from some distribution $p(a)$ with a variance of order 1 in $N$.

We can view this as an Anderson model [62] of $N$ sites with independent random on-site potentials and all-to-all random couplings on the order of $N^{-\gamma/2}$. From Fermi's golden rule, we can determine that the tunneling rate from one of these sites is on the order of $N^{1-\gamma}$. Thus

the Thouless time, the time at which the system will escape a single site and random matrix statistics first appears, is

$$t_{\text{Th}} \sim N^{\gamma - 1}. \tag{33}$$

The ergodicity-breaking transition occurs at the point where the tunneling rate is the same order as the spectral width [32, 48, 49, 63], or, equivalently, when the Thouless time becomes of the same order as the inverse spectral width. The spectral width of the GUE matrix $V$ is on the order of $N^{(1-\gamma)/2}$ [22], while the spectral width of $A$ is of order 1. So the spectral width of the RP model is

$$w \sim \max(1, N^{(1-\gamma)/2}). \tag{34}$$

This indicates that the ergodicity-breaking transition occurs at $\gamma = 1$. For all $\gamma < 1$ the RP model will exhibit GUE statistics.

The localization transition of the model, on the other hand, can be determined from the Mott criterion [32, 49, 63, 64], i.e., when the number of sites in resonance with a given one becomes finite at large $N$. For the RP model this number of sites is on the order of $N^{1-\gamma/2}$, indicating that the localization transition occurs at $\gamma = 2$. Equivalently, we can understand the localization transition as occuring when the Thouless and Heisenberg times are of the same order. From our result for the spectral width we find that the Heisenberg time, which is the inverse mean level spacing, is

$$t_{\text{Heis}} \sim \min(N, N^{(1+\gamma)/2}). \tag{35}$$

The Thouless and Heisenberg times are of the same order when $\gamma = 2$, which matches our earlier reasoning for the localization transition through the Mott criterion. For $\gamma > 2$ the Thouless time is larger than the Heisenberg time. Since the SFF reaches its plateau value at the Heisenberg timescale, there is no chance for random-matrix-like ramp to appear. The RP model will behave as if $H = A$. That is, Poissonian statistics will emerge. This analysis informs our determination of which values of $\gamma$ lead to which phase in the schematics of Fig. 1.

It's clear from the above discussion that the region $1 < \gamma < 2$ is particularly interesting because random-matrix-like behavior should be found, but only after a long period of time has elapsed. This case in which the Thouless time is larger than the inverse spectral width but smaller than the Heisenberg time is the nonergodic extended phase [27]. In this region eigenstates are not localized, but they are not spread sufficiently to be ergodic. For these values of $\gamma$ the calculation of the SFF is nontrivial because the behavior of the SFF depends not only on $\gamma$ but also the timescale. Going forward we will assume that $\gamma > 1$.

As a warm-up we calculate the Fourier transform of the density of states $\overline{C_1}(t)$ to leading order. Due to the independence of the eigenvalues of $A$ we can simplify Eq. (20) as

$$\overline{C_1}(t) = \frac{e^{-\sigma t^2/2}}{\tau} \oint_{\mathcal{C}} \frac{dz}{2\pi i} e^{itz} [g_1(z, \tau)]^N, \tag{36}$$

$$g_1(z, \tau) = 1 + \tau \left\langle \frac{1}{z - a'} \right\rangle, \tag{37}$$

where $a'$ is any eigenvalue of $A$. $\overline{C_1}(t)$ contains information about the average density of states. Expanding with the binomial theorem, we find

$$\frac{1}{N} \overline{C_1}(t) = \frac{1}{N} e^{-\sigma t^2/2} \oint_{\mathcal{C}} \frac{dz}{2\pi i} e^{itz} \sum_{j=0}^{N} \binom{N}{j} (i\sigma t)^{j-1} \left\langle \frac{1}{z - a'} \right\rangle^j. \tag{38}$$

Note that we have divided by $N$ to probe the average density of states but we have not performed the unfolding procedure. We want to examine this quantity on the scale of the spectral

width, so we let $t$ be order 1. For $\gamma > 1$, $N\sigma$ goes to 0, meaning only the $j = 1$ term remains. So

$$\frac{1}{N}\overline{C_1}(t) = \oint_{\mathcal{C}} \frac{dz}{2\pi i} e^{itz} \left\langle \frac{1}{z-a'} \right\rangle = \left\langle e^{ita'} \right\rangle. \tag{39}$$

This means that the average level density is the same as the probability density of the eigenvalues of $A$ [19].

As discussed above, we can neglect the disconnected contribution[1] so the unfolded SFF at non-zero times is determined entirely by $\overline{C_2}(t)$. In the RP model, Eq. (23) simplifies as

$$\overline{C_2}(t) = -\frac{e^{-\sigma t^2}}{\tau^2} \oint_{\mathcal{C}} \frac{dz}{2\pi i} \oint_{\mathcal{C}} \frac{dz'}{2\pi i} e^{it(z-z')} \left[1 - \left(\frac{\tau}{z'-z-\tau}\right)^2\right] [g_2(z,\tau;z',-\tau)]^N, \tag{40}$$

$$\begin{aligned} g_2(z,\tau;z',-\tau) &= \left\langle \left(1 + \frac{\tau}{z-a'}\right)\left(1 - \frac{\tau}{z'-a'}\right) \right\rangle \\ &= 1 + \tau\left(1 - \frac{\tau}{z'-z}\right)\left(\left\langle \frac{1}{z-a'} \right\rangle - \left\langle \frac{1}{z'-a'} \right\rangle\right). \end{aligned} \tag{41}$$

Inserting Eq. (40) into Eq. (24) and neglecting the disconnected contribution, we obtain

$$C(t) = -\frac{e^{-\sigma t^2}}{N\tau^2} \oint_{\mathcal{C}} \frac{dz}{2\pi i} \oint_{\mathcal{C}} \frac{dz'}{2\pi i} e^{it(z-z')} \left\{\left[1 - \left(\frac{\tau}{z'-z-\tau}\right)^2\right] [g_2(z,\tau;z',-\tau)]^N\right\}. \tag{42}$$

We now make the change of variables

$$(z,z') = x \pm \frac{y}{2N} - i(q,q')0^+, \tag{43}$$

where $q, q' = \pm 1$, indicating which leg of the contour $\mathcal{C}$ the variables are on. Note that this definition of $y$ differs from that of Eq. (27) by a factor of $p(x)$, but it has the same scaling with $N$ and the argument for taking $y$ to be order 1 still applies. This change is made purely for convenience. With it we obtain

$$C(t) = -\frac{e^{-\sigma t^2}}{(N\tau)^2} \sum_{q,q'} qq' \int \frac{dx}{2\pi i} \int \frac{dy}{2\pi i} e^{iN^{-1}ty} \left[1 - \left(\frac{N\tau}{y - i(q-q')0^+ + N\tau}\right)^2\right] [g_2(z,\tau;z',-\tau)]^N. \tag{44}$$

## 4.1 The spectral form factor

Kunz and Shapiro [45] used the process outlined above to find the SFF when $\gamma = 2$ and $t \sim N$ (for which the Thouless and Heisenberg timescales coincide). They found that, if the coupling parameter is also unfolded by replacing $\lambda$ with $\Lambda/p(x)$, the two-level cluster function and the SFF become independent of the center energy $x$. Later Kravtsov et al. [27] generalized this result to all $\gamma > 1$ at the Thouless timescale. Additional timescales may be examined by rescaling time in this result. We first review this result and then present our direct calculation of the SFF at all timescales of interest. This calculation more closely parallels that of the SFF for the BRP model presented in section 5.

Making the change of variables in Eq. (43), we find that

$$\left\langle \frac{1}{z-a'} \right\rangle = \int da' \frac{p(a')}{x-a'} + i\pi q p(x) + O(N^{-1}), \tag{45}$$

---

[1]Strictly speaking, the argument above for neglecting the disconnected contribution to the SFF only applies for $T \neq 0$, i.e., $t/N \neq 0$. Nonetheless, we make the same approximation for times that scale as a smaller (but still non-zero) power of $N$ as well. We find good agreement between the resulting expressions and numerical calculations for all timescales of interest.

where the $\fint$ symbol indicates the principal value of the integral. Using this result in Eq. (41), we determine that

$$g_2(z,\tau;z',-\tau) = 1 + \tau\left[i\pi p(x)(q-q') + O(N^{-1})\right]\left(1 + \frac{N\tau}{y - i(q-q')0^+}\right). \tag{46}$$

We examine the Thouless timescale by rescaling time as

$$t = N^{\gamma-1}s, \tag{47}$$

where $s$ is independent of $N$. Using this we can rewrite the above equation as

$$g_2(z,\tau;z',-\tau) = 1 - N^{-1}\pi\lambda^2 sp(x)(q-q')\left(1 + \frac{i\lambda^2 s}{y - i(q-q')0^+}\right) + O(N^{-2}). \tag{48}$$

With this we determine that, to leading order in $N^{-1}$,

$$[g_2(z,\tau;z',-\tau)]^N = \exp\left[-\pi\lambda^2 sp(x)(q-q')\left(1 + \frac{i\lambda^2 s}{y - i(q-q')0^+}\right)\right]. \tag{49}$$

Inserting this result into Eq. (44) and then unfolding by setting

$$v = s/p(x), \tag{50}$$
$$\Lambda = \lambda p(x), \tag{51}$$

yields the result that the Thouless-timescale SFF is

$$K_{\text{Th}}^{(\gamma)}(v) = 1 + e^{-2\pi\Lambda^2 v - N^{\gamma-2}\Lambda^2 v^2}$$
$$\times\left[\frac{2I_1(\kappa v^{3/2})}{\kappa v^{3/2}} - \frac{1}{4\pi}\kappa v^{5/2}N^{\gamma-2}\int_0^\infty \frac{d\xi\,\xi}{\sqrt{\xi+1}}I_1(\kappa v^{3/2}\sqrt{\xi+1})e^{-N^{\gamma-2}\Lambda^2 v^2\xi}\right], \tag{52}$$

where $\kappa = \sqrt{8\pi\Lambda^4 N^{\gamma-2}}$ and $I_1(x)$ is the modified Bessel function of the first kind. See Ref. [27] for full details. Note that we have unfolded using the probability density instead of the level density, which differ by a factor of $N$. Eq. (52) may also be used to examine other timescales by rescaling $v$. However, we note that the intermediate step Eq. (49) only holds for $s \ll \sqrt{N}$, that is when $t \ll N^{\gamma-1/2}$. Even restricting ourselves to times not larger than the Heisenberg time ($t \gg N$), we see that this is not always the case.

We now present a direct calculation of the SFF for the RP model at all relevant timescales. We make the same change of variables $\{z,z'\} \rightarrow \{x,y\}$ shown in Eq. (43), but we will not set the $N$-dependence of $t$ yet. We can now write

$$C(t) = \int dx\, p(x)e^{-\sigma t^2}(S_1 + S_2), \tag{53}$$

$$S_1 = -\frac{1}{2\pi i p(x)(N\tau)^2}\sum_{q,q'}qq'\int \frac{dy}{2\pi i}e^{iN^{-1}ty}[g_2(z,\tau;z',-\tau)]^N, \tag{54}$$

$$S_2 = \frac{1}{2\pi i p(x)}\sum_{q,q'}qq'\int \frac{dy}{2\pi i}e^{iN^{-1}ty}\frac{[g_2(z,\tau;z',-\tau)]^N}{[y - i(q-q')0^+ + N\tau]^2}. \tag{55}$$

Instead of seeking an asymptotic form for $[g(z,\tau;z',-\tau)]^N$ as in Eq. (49), we write

$$[g_2(z,\tau;z',-\tau)]^N = \sum_{j=0}^N \binom{N}{j}\frac{\left\{N\tau^2\left[i\pi p(x)(q-q') + O(N^{-1})\right]\right\}^j}{\{1 + \tau[i\pi p(x)(q-q') + O(N^{-1})]\}^{j-N}[y - i(q-q')0^+]^j}$$
$$= \sum_{j=0}^N p_{\text{bi}}\left(N,j,i\pi\tau p(x)(q'-q) + O(N^{-1}\tau)\right)\left(\frac{N\tau}{y - i(q-q')0^+}\right)^j, \tag{56}$$

where

$$p_{\text{bi}}(N, j, x) = \binom{N}{j}(1-x)^{N-j}x^j \tag{57}$$

is the binomial probability density function. We take $i\tau(q'-q)$ to be positive because, when it is not, the integrals over $y$ in Eqs. (54)-(55) vanish due to a lack of singularities in the half of the complex plane in which the contour of integration may be closed. At this point we restrict ourselves to times not larger than the Heisenberg time, so $|\tau| = \sigma t \ll 1$. This is not a problem since we already know that the SFF goes to 1 for times larger than the Heisenberg time. With this restriction, we see by inspection of the first line of Eq. (56) that we can neglect the $O(N^{-1})$ terms when $j \ll N$. The probability density $p_{\text{bi}}(N, j, x)$ is peaked near its mean at $j = Nx$ with a variance of $Nx(1-x)$. Far from the peak the probability density is exponentially small in $N$. So only the $j \sim N|\tau|$ terms will contribute in Eq. (56). This means that the $j \sim N$ terms will not contribute and we can therefore neglect the $O(N^{-1})$ terms. With these considerations we can write, for positive $i\tau(q'-q)$,

$$[g_2(z, \tau; z', -\tau)]^N = \sum_{j=0}^{N} p_{\text{bi}}\left(N, j, i\pi\tau p(x)(q'-q)\right)\left(\frac{N\tau}{y - i(q-q')0^+}\right)^j. \tag{58}$$

Inserting this into Eq. (54) we find

$$S_1 = \frac{1}{N}\sum_q\sum_{j=1}^{N}\binom{N}{j}\frac{(2\pi i N\tau^2 p(x))^{j-1}}{(1+2\pi i\tau q p(x))^{j-N}}\int\frac{dy}{2\pi i}\frac{e^{iN^{-1}tqy}}{(y-i0^+)^j}, \tag{59}$$

where we have made the change of variable $y \to qy$. Note that the $j = 0$ term vanishes due to the integrand having no singularities. For the remaining terms the integral over $y$ vanishes when $tq < 0$, so we can remove the sum over $q$ by making the replacement $tq \to |t|$. Performing the integral over $y$, we find that

$$S_1 = \sum_{j=0}^{N-1}\frac{(|\tau||t|)^j}{(j+1)!}p_{\text{bi}}(N-1, j, 2\pi|\tau|p(x)). \tag{60}$$

We now turn our attention to $S_2$. Inserting Eq. (56) into Eq. (55), we find

$$S_2 = \frac{i}{2\pi p(x)}\sum_q\sum_{j=1}^{N}\binom{N}{j}\frac{(2\pi i N\tau^2 p(x))^j}{(1+2\pi i\tau q p(x))^{j-N}}\int\frac{dy}{2\pi i}\frac{e^{iN^{-1}tqy}}{(y+N\tau q)^2(y-i0^+)^j}, \tag{61}$$

where we have again made the change of variable $y \to qy$. The $j = 0$ term vanishes because the integrand has no singularities in the half of the complex plane in which the contour of integration may be closed. For the remaining terms the integral over $y$ vanishes when $tq < 0$, so we again remove the sum over $q$ by making the replacement $tq \to |t|$. Performing the integral over $y$, we find that

$$S_2 = -e^{\sigma t^2}\sum_{j=0}^{N-1}p_{\text{bi}}(N-1, j, 2\pi|\tau|p(x))\left[\tilde{\Gamma}(j+2, \sigma t^2) - \frac{\sigma t^2}{j+1}\tilde{\Gamma}(j+1, \sigma t^2)\right], \tag{62}$$

where

$$\tilde{\Gamma}(j+1, z) = \frac{1}{j!}\int_z^{\infty}d\xi\,\xi^j e^{-\xi} = e^{-z}\sum_{k=0}^{j}\frac{z^k}{k!} \tag{63}$$

is the regularized upper incomplete gamma function. Note that $\tilde{\Gamma}$ satisfies the recurrence relation

$$\tilde{\Gamma}(j+1, z) = \tilde{\Gamma}(j, z) + \frac{z^j e^{-z}}{j!}. \tag{64}$$

We take our results for $S_1$ and $S_2$ in Eqs. (60) and (62) and insert them into Eq. (53), then extract the SFF using Eq. (29). Then unfolding time by setting

$$u = |t|/p(x) = N|T|,\qquad(65)$$

and unfolding $\lambda$ using Eq. (51), we obtain the SFF

$$K^{(\gamma)}(u/N) = 1 + \sum_{j=0}^{N-1} p_{\text{bi}}(N-1, j, 2\pi N^{-\gamma}\Lambda^2 u)$$

$$\times \left[ \frac{e^{-N^{-\gamma}\Lambda^2 u^2}}{(j+1)!} \left(N^{-\gamma}\Lambda^2 u^2\right)^j - \tilde{\Gamma}\left(j+2, N^{-\gamma}\Lambda^2 u^2\right) + \frac{N^{-\gamma}\Lambda^2 u^2}{j+1} \tilde{\Gamma}\left(j+1, N^{-\gamma}\Lambda^2 u^2\right) \right].\qquad(66)$$

With this result we have generalized the Thouless-timescale result of Eq. (52) to all times larger than the inverse spectral width. Although it was derived considering only times not larger than the Heisenberg time, it continues to hold for later times, as we will discuss in the following section.

Fig. 4 plots Eq. (66) for $\gamma = 1.6$ and various values of the coupling parameter $\Lambda$ and compares to numerical results obtained through exact diagonalization. There is good agreement between theory and numerics at not-too-early times (note in particular that the agreement is already good by the Thouless timescale). The discrepancy at early times which are still larger than the inverse spectral width is a result of calculating the numerical SFF using only the eigenvalues within a finite window in order to perform the unfolding procedure. At times earlier than this inverse window width the SFF probes correlations between eigenvalues at separations larger than the width of the window, but these eigenvalues are cut off in the numerical unfolding procedure. We observe that the SFF decays from the Poissonian result to the GUE SFF over time, with the rate of decay controlled by $\Lambda$. As we vary $\Lambda$ we can see the SFF moving between the behaviors we predicted schematically in Fig. 1. For small values of $\Lambda$ we see that the SFF reaches its plateau at times greater (but not much greater) than the Heisenberg time. The agreement between theory and numerics is still very good at these late times.

If we examine the Thouless timescale by setting $u = N^{\gamma-1}v$ in Eq. (66) we can write

$$K_{\text{Th}}^{(\gamma)}(v) = K^{(\gamma)}(N^{\gamma-2}v) = 1 + e^{-2\pi\Lambda^2 v - N^{\gamma-2}\Lambda^2 v^2} \frac{2I_1(\kappa v^{3/2})}{\kappa v^{3/2}} - B_{\text{Th}}^{(\gamma)}(v),\qquad(67)$$

$$B_{\text{Th}}^{(\gamma)}(v) = e^{-2\pi\Lambda^2 v} \sum_{j=0}^{\infty} \frac{(2\pi\Lambda^2 v)^j}{j!} \left( \tilde{\Gamma}(j+2, \Lambda^2 N^{\gamma-2}v^2) + \frac{\Lambda^2 N^{\gamma-2}v^2}{j+1} \tilde{\Gamma}(j+1, \Lambda^2 N^{\gamma-2}v^2) \right),\quad(68)$$

where we have discarded some vanishing terms and used the fact that the modified Bessel function may be written as the infinite series

$$I_1(x) = \sum_{j=0}^{\infty} \frac{(x/2)^{2j+1}}{j!(j+1)!}.\qquad(69)$$

Writing the regularized upper incomplete gamma functions in the integral form shown in Eq. (63), using Eq. (69), and making the change of variable $\xi \to \sqrt{\xi + 1}$, we recover the Thouless-timescale SFF shown in Eq. (52).

## 4.2 The infinite matrix size limit

We can now determine the large $N$ limit of the SFF at different timescales. We begin by examining the Thouless timescale where $u \sim N^{\gamma-1}$. When $\gamma > 2$ the regularized incomplete gamma functions in Eq. (68) vanish, so

$$B_{\text{Th}}^{(\gamma)}(v) = 0.\qquad(70)$$
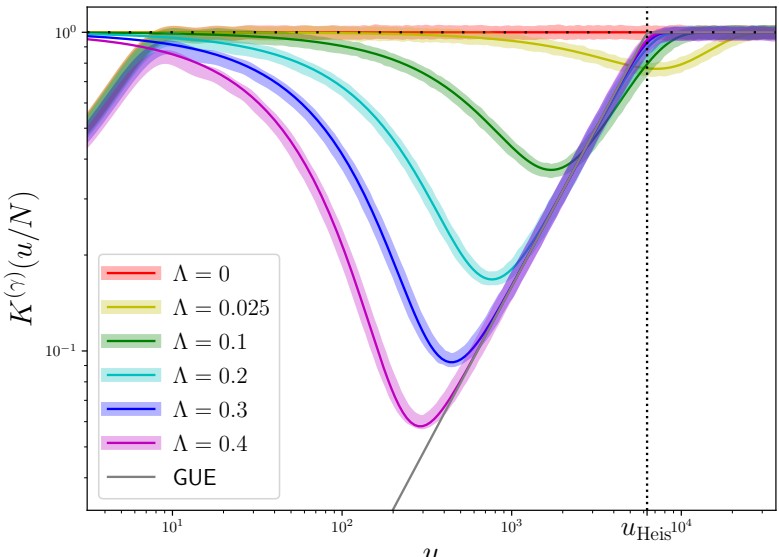

Figure 4: The SFF of the RP model when $\gamma = 1.6$ for several values of the coupling parameter $\Lambda$. The dark thin lines show the analytical results while the lighter, thicker lines show numerical results. The dotted line indicates the Heisenberg time. In the numerics the diagonal elements of $A$ are drawn from the Gaussian distribution with 0 mean and a variance of 1. The numerics are obtained through exact diagonalization of size $N = 1000$ random matrices and are averaged over $10^5$ realizations. The numerics are restricted to eigenvalues within the window [-1,1].

Inserting this result in Eq. (67) and noting that the second term of the SFF vanishes for large $N$, we find that the SFF is

$$K_{\text{Th}}^{(\gamma)}(v) = 1, \tag{71}$$

indicating localized behavior.

When $\gamma < 2$ the regularized incomplete gamma functions become complete and go to 1. Resumming then gives

$$B_{\text{Th}}^{(\gamma)}(v) = 1 - \frac{T}{2\pi}\left(1 - e^{-2\pi\Lambda^2 v}\right), \tag{72}$$

where $T = t/\rho(x) = N^{\gamma-2}v$ is the fully unfolded (including all factors of $N$) time. We again insert this in Eq. (67). We note that the second term of the SFF goes to $e^{-2\pi\Lambda^2 v}$, so the SFF for $1 < \gamma < 2$ is

$$K_{\text{Th}}^{(\gamma)}(v) = e^{-2\pi\Lambda^2 v} + \frac{T}{2\pi}\left(1 - e^{-2\pi\Lambda^2 v}\right). \tag{73}$$

The second term is subleading, but we include it to make the crossover to GUE behavior for large $\Lambda$ apparent. $K_{\text{Th}}^{(2)}(v)$ does not have a simple limit.

Pulling together all the results for the Thouless timescale SFF at different values of $\gamma$ we have

$$K_{\text{Th}}^{(\gamma)}(v) = \begin{cases} 1, & \gamma > 2, \\ 1 + e^{-\Lambda^2 v(v+2\pi)}\frac{2I_1(\kappa v^{3/2})}{\kappa v^{3/2}} - B_{\text{Th}}^{(2)}(v), & \gamma = 2, \\ e^{-2\pi\Lambda^2 v} + \frac{T}{2\pi}\left(1 - e^{-2\pi\Lambda^2 v}\right), & 1 < \gamma < 2. \end{cases} \tag{74}$$

When $1 < \gamma \leq 2$ the behavior of the SFF is dependent on $\Lambda$. It moves to the Poissonian result for small $\Lambda$ and the GUE result for large $\Lambda$. This can be seen explicitly in the above equation when $1 < \gamma < 2$.

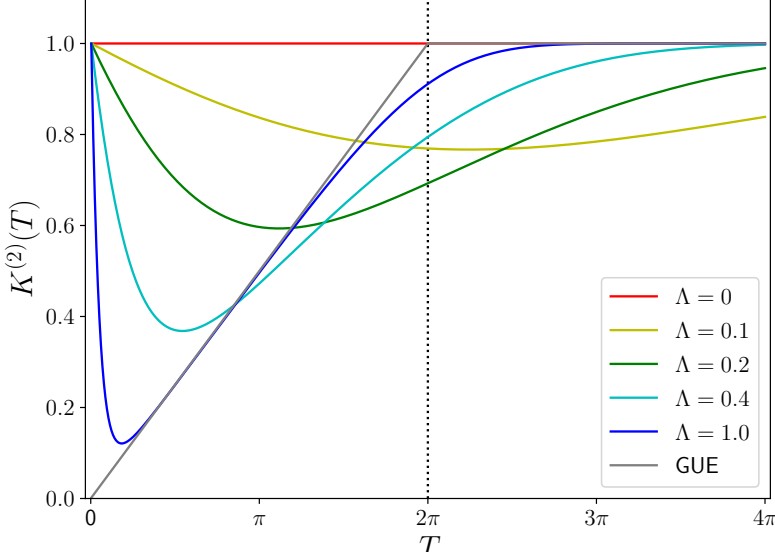

Figure 5: The SFF for the RP model when the Thouless and Heisenberg timescales coincide for various values of the coupling parameter $\Lambda$. The dotted vertical line indicates the Heisenberg time.

It is not surprising that the SFF does not have a simple limit when $\gamma = 2$ since, in this case, the Thouless and Heisenberg timescales are the same. By plotting this case, shown in Fig. 5, we can observe that there is still a crossover from the Poissonian to the GUE result controlled by $\Lambda$. An interesting feature of this crossover is that, for finite $\Lambda$, the SFF saturates at times later than the Heisenberg time. There is a trade-off: increasing the level repulsion at large energy separations decreases the level repulsion at small separations and vice versa. This phenomenon can be shown through the fact that the area between the SFF and its plateau value is independent of the coupling strength for any nonzero coupling. This property is demonstrated in the Appendix.

We can now examine the large $N$ limit of the SFF at other timescales. For convenience we rewrite the SFF of Eq. (66) as

$$
K^{(\gamma)}(u/N) = 1 + e^{-N^{-\gamma}\Lambda^2 u^2} \sum_{j=0}^{N-1} \frac{(N^{-\gamma}\Lambda^2 u^2)^j}{(j+1)!} p_{\text{bi}}(N-1, j, 2\pi N^{-\gamma}\Lambda^2 u) - B^{(\gamma)}(u), \tag{75}
$$

$$
B^{(\gamma)}(u) = \sum_{j=0}^{N-1} p_{\text{bi}}(N-1, j, 2\pi N^{-\gamma}\Lambda^2 u) \left[ \tilde{\Gamma}\left(j+2, N^{-\gamma}\Lambda^2 u^2\right) - \frac{N^{-\gamma}\Lambda^2 u^2}{j+1} \tilde{\Gamma}\left(j+1, N^{-\gamma}\Lambda^2 u^2\right) \right]. \tag{76}
$$

The regularized upper incomplete gamma function $\tilde{\Gamma}(j, x)$ is the cumulative distribution function for a Poissonian random variable with parameter $x$. This means that these functions in Eq. (76) have a crossover from 0 to 1, where the location of the crossover is at approximately $j = N^{-\gamma}\Lambda^2 u^2$ and the width of the crossover is on the order of $N^{-\gamma/2}\Lambda u$. Meanwhile the binomial probability density function in Eq. (76) has a mean and variance of approximately $2\pi N^{1-\gamma}\Lambda^2 u$ for large $N$ and $u$ not larger than order $N$.

In the following considerations we will assume that the crossover point of the upper incomplete gamma functions and the mean of the binomial distribution are much larger than the crossover width and the binomial distribution width. Fortunately, for the timescales we are considering, this is only violated when $u \sim N$ and $\gamma = 2$. This is the case where the Thouless and Heisenberg timescales coincide, which we have already considered above.

The binomial distribution is exponentially small far from its mean, so only the terms with $j$ close to $2\pi N^{1-\gamma}\Lambda^2 u$ may not vanish. If the mean of the binomial distribution is less than the crossover point of the regularized upper incomplete gamma functions, which occurs when $u > 2\pi N$ and $\gamma < 2$, we see from Eq. (63) that those functions will be exponentially small and can therefore be replaced with 0. If the mean of the binomial distribution is greater than the crossover point, which occurs when $u < 2\pi N$, they can be replaced with 1. We can also observe that when $\gamma > 2$, the regularized incomplete gamma functions will become complete and approach 1. With these considerations we find, after performing the sum over $j$, that for $u \ggg N$

$$B^{(\gamma)}(u) = \begin{cases} 1 - \frac{u}{2\pi N}\left[1 - (1 - 2\pi N^{-\gamma}\Lambda^2 u)^N\right], & u < 2\pi N \text{ or } \gamma > 2, \\ 0, & u > 2\pi N \text{ and } \gamma < 2. \end{cases} \tag{77}$$

We now seek to find the large $N$ limit of the SFF for times smaller than the Thouless time and not larger than the Heisenberg time, that is when $u \ll t_{\text{Th}} \sim N^{\gamma-1}$ and $u \ggg t_{\text{Heis}} \sim N$. Under these conditions we find that $e^{-N^{-\gamma}\Lambda^2 u^2} \to 1$ and the mean and variance of the binomial density in Eq. (75) go to 0, leaving only the $j = 0$ contribution nonvanishing in the second term. We also find that $(1 - 2\pi N^{-\gamma}\Lambda^2 u)^N \to 1$. Using these limits in Eqs. (75) and (77) we find that the SFF is

$$K^{(\gamma)}(T) = 1. \tag{78}$$

This matches the result for the SFF at times much later than the Heisenberg time, so this result is valid for all times smaller than the Thouless time but larger than the inverse spectral width.

At these times the SFF indicates Poissonian statistics. This can also be determined directly from Eqs. (19) and (22). When $\sigma t^2$ and $N\tau$ are both less than order 1, which is the case for times smaller than the Thouless time and not larger than the Heisenberg time, we find in the $N \to \infty$ limit that

$$\overline{C_1}(t) = \left\langle \sum_k e^{ita_k} \right\rangle, \qquad \overline{C_2}(t) = \left\langle \sum_{k \neq l} e^{it(a_k - a_l)} \right\rangle, \tag{79}$$

indicating that the SFF is simply that of $A$. Since the eigenvalues of $A$ are uncorrelated, Poissonian statistics are found.

We now seek to find the large $N$ limit of the SFF for times larger than the Thouless time but not larger than the Heisenberg time, that is when $u \gg t_{\text{Th}} \sim N^{\gamma-1}$ and $u \ggg t_{\text{Heis}} \sim N$. We note that the maximum value of the binomial probability density is on the order of the inverse of its standard deviation. This tells us that

$$\sum_{j=0}^{N-1} \frac{(N^{-\gamma}\Lambda^2 u^2)^j}{(j+1)!} p_{\text{bi}}(N-1, j, 2\pi N^{-\gamma}\Lambda^2 u) \ll \sum_{j=0}^{\infty} \frac{(N^{-\gamma}\Lambda^2 u^2)^j}{j!} = e^{N^{-\gamma}\Lambda^2 u^2}. \tag{80}$$

Therefore the second term in Eq. (75) vanishes. We also note that $(1 - 2\pi N^{-\gamma}\Lambda^2 u)^N \to 0$. With these considerations we find that the SFF is

$$K^{(\gamma)}(T) = \min\left(\frac{T}{2\pi}, 1\right). \tag{81}$$

This result for the SFF is equal to 1 for all times later than the Heisenberg time, so it is valid for all times larger than the Thouless time and inverse spectral width. We have recovered the GUE SFF. We know that GUE statistics hold for $\gamma < 1$, so we can conclude that this result is valid for all values of $\gamma$.

Finally we examine the large $N$ behavior of the SFF at times larger than the Heisenberg time, where $u \gg t_{\text{Heis}} \sim N$. Although we derived the SFF assuming earlier times, the numerical

results shown in Fig. 4 indicate that our result may also capture the plateau behavior. The way to determine the large $N$ behavior at these late times depends on $\gamma$. When $\gamma > 2\log_N u$ we see that the mean and variance of the binomial distribution in Eqs. (75)-(76) vanish, meaning the distribution goes to $\delta_{j0}$. Additionally, the regularized incomplete gamma functions become complete in this case and go to 1. Making these replacements we see that the SFF goes to 1. When $\log_N u < \gamma < 2\log_N u$ the crossovers of the regularized incomplete gamma functions are much larger than the mean of the binomial distribution, so the gamma functions may be taken to be 0, meaning $B^{(\gamma)}(u)$ vanishes. Eq. (80) holds in this case, so the second term in Eq. (75) also vanishes, showing that the SFF goes to 1. In the last case, where $1 < \gamma < \log_N u$, the function $p_{\mathrm{bi}}(N-1, j, 2\pi N^{-\gamma}\Lambda^2 u)$ can no longer be considered a probability distribution since its parameter is larger than 1, causing it to take on negative values for odd $j$. Because $N^{-\gamma}\Lambda^2 u^2 \gg N$ in this case, we see from Eq. (63) that the regularized incomplete gamma functions are dominated by their highest order terms. Taking the summand in Eq. (66) to leading order we find that the SFF is

$$K^{(\gamma)}(u/N) = 1 - 2e^{-N^{-\gamma}\Lambda^2 u^2} \sum_{j=0}^{N-1} p_{\mathrm{bi}}(N-1, j, 2\pi N^{-\gamma}\Lambda^2 u)\frac{j}{(j+1)!}(N^{-\gamma}\Lambda^2 u^2)^{j-1}. \tag{82}$$

The sum in the above expression is also dominated by its highest order term, which grows slower than the exponential factor vanishes. Again the SFF goes to 1. These considerations show that the plateau behavior is recovered for all times much larger than the Heisenberg time. This accounts for the good agreement between the theory and numerics at these late times.

We can visualize the results of Ref. [27] and the new results found here with a diagram for the behavior of the SFF at different timescales and values of $\gamma$, shown in Fig. 6. The SFF is 1 before the Thouless timescale and equal to the GUE SFF after. At the Thouless timescale, when it is not greater than the Heisenberg time, there is a crossover between these two behaviors. At times larger than the Heisenberg time the SFF is simply 1.

## 5 The block Rosenzweig-Porter model

Motivated by our recent results for the early time ramp of the SFF for a quantum spin glass [46], we construct a minimal quantum model which can also exhibit glassy behavior. We create this model by generalizing the RP model examined in the previous section. The diagonal matrix $A$ in Eq. (32) is redefined as a block-diagonal matrix with $P$ independent GUE matrices of size $M = N/P$. Each block in this matrix corresponds to a single sector and the fact that the block is a GUE matrix means that the system is ergodic within that sector. The Hamiltonian is the sum of this block-diagonal matrix and a size $N$ GUE matrix which couples the sectors.

More formally, the Hamiltonian matrix for the BRP model has the form of Eq. (8), but now with the condition that the eigenvalues of $A$ are those of $P$ independent size $M$ GUE matrices drawn from the distribution

$$p\left(A^{(i)}\right) \sim \exp\left[-\frac{M}{2}\mathrm{tr}\left(A^{(i)}\right)^2\right]. \tag{83}$$

This means that the spectral width of $A$ is order 1 and, if $M$ is large, the eigenvalues of $A$ cannot have magnitude greater than 2 [4]. We can consider $A$ to be block diagonal such that

$$A = \bigoplus_{i=1}^{P} A^{(i)}. \tag{84}$$

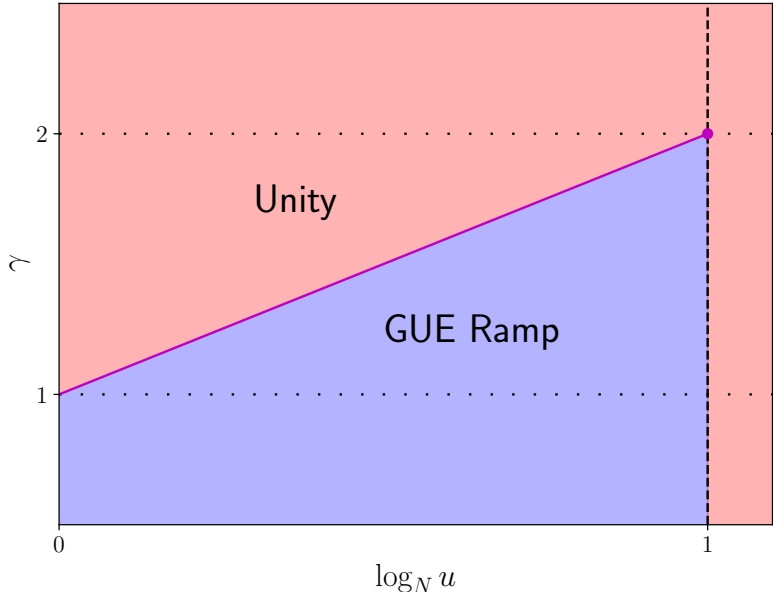

Figure 6: The SFF of the RP model at different timescales and values of the coupling parameter $\gamma$. The magenta dot indicates the point where the Thouless and Heisenberg timescales coincide, where a closed form expression for the SFF is not known. The dotted lines indicate the values of $\gamma$ at which there is a phase transition. The dashed line indicates the Heisenberg time.

Similar to our analysis of the RP model, we can use the Fermi golden rule to determine the Thouless timescale and determine the locations of the transitions by comparing to the other cardinal timescales. We are interested in the escape rate from a particular site to sites outside that starting site's sector. The number of such sites to escape to is of order $N$, the coupling is of order $N^{-\gamma/2}$, and the spectral width of $A$ is order 1. It follows that the RP results for the Thouless time, the spectral width, and the Heisenberg time, shown in Eqs. (33)-(35) respectively, are valid for the BRP model also.

The localization transition occurs when the Thouless and Heisenberg times are of the same order, which is when $\gamma = 2$. Notice that the block-diagonal structure of $A$ leads to a localized phase in the BRP model which is distinct from that of the RP model. Within this phase the system will be confined to the subspaces corresponding the the blocks of $A$ instead of individual states. The transition to full GUE statistics occurs when the Thouless time is of the same order as the inverse spectral width. This occurs when $\gamma = 1$.

For the BRP model there is another timescale of interest—the block Heisenberg time $t_{\mathrm{Heis,Bl}} = t_{\mathrm{Heis}}/P$. This is the inverse mean level spacing for a single block of $A$. In the large $\gamma$ limit where the coupling between the blocks vanishes, $t_{\mathrm{Heis,Bl}}$ is the time at which the SFF will reach its plateau value. Comparing the Thouless time to the block Heisenberg time, we see that there will be another transition at $\gamma = 1 + d$, where $d = \log_N M$, which is when these timescales are of the same order. This analysis informs our determination of which values of $\gamma$ lead to which phase in the schematics of Fig. 2.

Two of the transitions we have discussed have an important interpretation in terms of the size of the support set of the energy eigenstates. Starting with an unperturbed energy eigenstate of $A$, which is spread over $M$ sites in a single block, the perturbation from the $V$ matrix causes each of those $M$ sites to hybridize with other sites within an energy interval on the order of $E_{\mathrm{Th}} \sim t_{\mathrm{Th}}^{-1}$. Assuming this interval is not smaller than the mean level spacing $\delta = w/N$, the number of sites in this interval is $E_{\mathrm{Th}}/\delta$. This means that the size of the support

set for the energy eigenstates is on the order of $ME_{\text{Th}}/\delta = N^{2-\gamma+d}$. This is of order $M$ when $\gamma = 2$, indicating the localization transition. For larger $\gamma$, $E_{\text{Th}}$ becomes smaller than $\delta$ and no hybridization occurs. The size of the support set is of order $N$ when $\gamma = 1 + d$, indicating delocalization over the entire Hilbert space. It is an interesting feature of the BRP model that this transition is, in general, separate from the transition to full GUE spectral statistics at $\gamma = 1$.

We note that if the block size $M$ is of order 1, the BRP model becomes equivalent to the RP model. In this case the full eigenstate delocalization transition occurs at $\gamma = 1$ as it does for the RP model. In terms of the SFF, the eigenvalues of $A$ are uncorrelated on timescales larger than the block Heisenberg time. When $M$ is of order 1 the block Heisenberg time is of the same order as the inverse spectral width. Since the eigenvalues of $A$ are uncorrelated for the relevant timescales, the BRP model reduces to the RP model. Going forward we will consider $M$ to be larger than order 1.

We may now turn our attention to the calculation of the SFF for the BRP model. We will once again assume in our calculations that $\gamma > 1$. With the additional information about the structure of $A$ we can simplify Eqs. (20) and (23) for $\overline{C_1}(t)$ and $\overline{C_2}(t)$ respectively. Each eigenvalue of $A$ will be correlated only with the eigenvalues from the same block so

$$\overline{C_1}(t) = \frac{e^{-\sigma t^2/2}}{\tau} \oint_{\mathcal{C}} \frac{dz}{2\pi i} e^{itz} [Z_1(z,\tau)]^P \,, \tag{85}$$

$$Z_1(z,\tau) = \left\langle \frac{\det(z + \tau - A')}{\det(z - A')} \right\rangle \,, \tag{86}$$

and

$$\overline{C_2}(t) = -\frac{e^{-\sigma t^2}}{\tau^2} \oint \frac{dz}{2\pi i} \oint \frac{dz'}{2\pi i} e^{it(z-z')} \left[1 - \left(\frac{\tau}{z' - z - \tau}\right)^2\right] [Z_2(z,\tau;z',-\tau)]^P \,, \tag{87}$$

$$Z_2(z,\tau;z',-\tau) = \left\langle \frac{\det(z + \tau - A')\det(z' - \tau - A')}{\det(z - A')\det(z' - A')} \right\rangle \,, \tag{88}$$

where $A'$ is any of the $A^{(i)}$. The functions $Z_1$ and $Z_2$ are actually generating functions. The ensemble averaged Green's function for a size $M$ GUE matrix is

$$\langle G(z)\rangle = \left\langle \text{tr}\, \frac{1}{z - A'} \right\rangle = \left[\frac{\partial}{\partial \tau} Z_1(z,\tau)\right]_{\tau=0} \,, \tag{89}$$

and the correlator of two Green's functions is

$$\langle G(z)G(z')\rangle = \left\langle \text{tr}\, \frac{1}{z - A'} \text{tr}\, \frac{1}{z' - A'} \right\rangle = \left[\frac{\partial^2}{\partial \tau \partial \tau'} Z_2(z,\tau;z',\tau')\right]_{\tau,\tau'=0} \,. \tag{90}$$

We now consider whether the condition on the average level density which we derived in Eq. (39) for the RP model holds for the BRP model also. From Eq. (86) we find that

$$Z_1(z,\tau) = 1 + M\tau \left\langle \frac{1}{z - a'} \right\rangle + O(\tau^2) \,. \tag{91}$$

Using this in Eq. (85) and dividing by $N$ while letting $t$ be order 1, the same order as the inverse spectral width, we find that Eq. (39) does indeed hold for the BRP model. This means that, when $\gamma > 1$, the average level density is equal to the probability density for the eigenvalues of $A$, which is the average level density for size $M$ GUE matrices. For infinitely large $M$ this is

$$p(a) = \begin{cases} \frac{1}{2\pi}\sqrt{4 - a^2}, & |a| \leq 2, \\ 0, & |a| > 2. \end{cases} \tag{92}$$

In this section we will make the change of variables

$$(z, z') = x \pm \frac{y}{2N} + i(q, q')0^+ . \tag{93}$$

This differs from the change of variables of Eq. (43) only in the signs of $q$ and $q'$, which is done purely for convenience. With this change of variables we see that

$$C(t) = -\frac{e^{-\sigma t^2}}{(N\tau)^2} \sum_{qq'} qq' \int \frac{dx}{2\pi i} \int \frac{dy}{2\pi i} e^{iN^{-1}ty} \left[ 1 - \left( \frac{N\tau}{y + N\tau + i(q - q')0^+} \right)^2 \right] [Z_2(z, \tau; z', -\tau)]^P . \tag{94}$$

As we did for the calculation of the SFF for the RP model, we are neglecting the contribution from the disconnected part of the two-level correlation function under the assumption that correlations between energy levels with separations much larger than the mean level spacing will vanish.

## 5.1 The two-point generating function

In this section we calculate the two-point generating function

$$Z_2(z, \tau; z', -\tau) = \left\langle \frac{\det(z + \tau - A') \det(z' - \tau - A')}{\det(z - A') \det(z' - A')} \right\rangle , \tag{95}$$

where $A'$ is a GUE matrix of size $M$ drawn from the distribution

$$p(A') \sim \exp\left( -\frac{M}{2} \mathrm{tr} A'^2 \right) . \tag{96}$$

We begin by writing the determinants in the numerator as fermionic integrals and the determinants in the denominator as bosonic integrals. Doing so, we obtain

$$
\begin{aligned}
&Z_2(z, \tau; z', -\tau) \\
&= \pi^{-2M} \left\langle \int D(\bar\psi, \psi) \exp \sum_\zeta \sum_\alpha \left\{ \sum_i \bar\psi^\alpha_{\zeta i} A'_{ii} \psi^\alpha_{\zeta i} + \sum_{i>j} \left( \bar\psi^\alpha_{\zeta i} \psi^\alpha_{\zeta j} + \bar\psi^\alpha_{\zeta j} \psi^\alpha_{\zeta i} \right) \Re(A'_{ij}) \right. \right. \\
&\left. \left. + i \sum_{i>j} \left( \bar\psi^\alpha_{\zeta i} \psi^\alpha_{\zeta j} - \bar\psi^\alpha_{\zeta j} \psi^\alpha_{\zeta i} \right) \Im(A'_{ij}) - \left[ \left( z^{(\alpha)} - (-1)^\alpha \tau \right) \delta_{\zeta F} + z^{(\alpha)} \delta_{\zeta B} \right] \sum_i \bar\psi^\alpha_{\zeta i} \psi^\alpha_{\zeta i} \right\} \right\rangle , \tag{97}
\end{aligned}
$$

where $\alpha = 1, 2$, $(z^{(1)}, z^{(2)}) = (z, z')$, and $\zeta = B, F$, indicating bosonic and fermionic fields respectively.

Performing the GUE average over $A'$, we obtain the following mean-field sigma model:

$$
\begin{aligned}
Z_2(z, \tau; z', -\tau) = \pi^{-2M} \int D(\bar\psi, \psi) \exp \left\{ \frac{1}{2M} \mathrm{str} \tilde\Omega \right. \\
\left. - \sum_\zeta \sum_\alpha \sum_i \left[ \left( z^{(\alpha)} - (-1)^\alpha \tau \right) \delta_{\zeta F} + z^{(\alpha)} \delta_{\zeta B} \right] \bar\psi^\alpha_{\zeta i} \psi^\alpha_{\zeta i} \right\} , \tag{98}
\end{aligned}
$$

$$\tilde\Omega = \begin{pmatrix} \tilde\Omega_{BB} & \tilde\Omega_{BF} \\ \tilde\Omega_{FB} & \tilde\Omega_{FF} \end{pmatrix} , \qquad \tilde\Omega_{\zeta\zeta'} = \begin{pmatrix} \sum_i \psi^1_{\zeta i} \psi^1_{\zeta' i} & \sum_i \psi^1_{\zeta i} \psi^2_{\zeta' i} \\ \sum_i \psi^2_{\zeta i} \psi^1_{\zeta' i} & \sum_i \psi^2_{\zeta i} \psi^2_{\zeta' i} \end{pmatrix} , \tag{99}$$

where $\tilde{\Omega}$ is a $4 \times 4$ matrix and $\operatorname{str}\tilde{\Omega} = \operatorname{tr}\tilde{\Omega}_{BB} - \operatorname{tr}\tilde{\Omega}_{FF}$ is the supertrace. We perform a Hubbard-Stratonovich transformation to remove the quartic terms in the action by inserting the unity

$$1 = \frac{1}{4\pi^4} \int d\Omega \exp\left(-\frac{1}{2M}\operatorname{str}\tilde{\Omega}^2 - \bar{\Psi}\Omega\Psi - \frac{M}{2}\operatorname{str}\Omega^2\right), \tag{100}$$

$$\Omega = \begin{pmatrix} \Omega_{BB} & \Omega_{BF} \\ \Omega_{FB} & i\Omega_{FF} \end{pmatrix}, \qquad \Omega_{\zeta\zeta'} = \begin{pmatrix} \Omega_{\zeta\zeta'}^{11} & \Omega_{\zeta\zeta'}^{12} \\ \Omega_{\zeta\zeta'}^{21} & \Omega_{\zeta\zeta'}^{22} \end{pmatrix}, \qquad \Psi = \begin{pmatrix} \psi_B^1 \\ \psi_B^2 \\ \psi_F^1 \\ \psi_F^2 \end{pmatrix}, \tag{101}$$

where $d\Omega = d\Omega_{BB}d\Omega_{FF}d(\Omega_{BF}, \Omega_{FB})$. The field $\Omega_{\zeta\zeta'}^{\alpha\alpha'}$ is bosonic when $\zeta = \zeta'$ and fermionic when $\zeta \neq \zeta'$. Inserting this unity and integrating out the $\Psi$ field, we find

$$Z_2(z, \tau; z', -\tau) = \frac{1}{4\pi^4} \int d\Omega [\operatorname{sdet}(\Omega + \mathcal{E})]^{-M} e^{-M\operatorname{str}\Omega^2/2}, \tag{102}$$

$$\mathcal{E} = \operatorname{diag}(z^{(1)}, z^{(2)}, z^{(1)} + \tau, z^{(2)} - \tau), \tag{103}$$

where $\operatorname{sdet}\Omega = \det(\Omega_{BB} - \Omega_{BF}\Omega_{FF}^{-1}\Omega_{FB})(\det\Omega_{FF})^{-1} = \det\Omega_{BB}[\det(\Omega_{FF} - \Omega_{FB}\Omega_{BB}^{-1}\Omega_{BF})]^{-1}$ is the superdeterminant.

We now make the change of variable $\Omega \to U\omega U^\dagger - \mathcal{E}$, where $U$ is an element of the superunitary group $U(2|2)$ and $\omega = \operatorname{diag}(\omega_B^1, \omega_B^2, i\omega_F^1, i\omega_F^2)$. The Jacobian of this change of variables is the supersymmetric generalization of the Vandermonde determinant [65] $\Delta_s(\Omega) = \det(1/(\omega_{B,i} - i\omega_{F,j}))$, where $\omega_{B,i}$ are the Bose-Bose eigenvalues and $i\omega_{F,j}$ are the Fermi-Fermi eigenvalues. We can now write

$$Z_2(z, \tau; z', -\tau) = \frac{1}{4\pi^4} \int d(\omega - \mathcal{E})\Delta_s^2(\omega)(\operatorname{sdet}\omega)^{-M} \int dU e^{-M(U\omega U^\dagger - \mathcal{E})^2/2}, \tag{104}$$

where $dU$ is the Haar measure of the superunitary group $U(2|2)$.

We can now use the supersymmetric generalization of the Itzykson-Zuber integral [65]:

$$\int dU e^{-M\operatorname{str}(U\omega U^\dagger - \beta)^2} \sim M^k [\Delta_s(\omega)\Delta_s(\eta)]^{-1} e^{-M\operatorname{str}(\omega-\eta)^2}, \tag{105}$$

where $dU$ is the Haar measure of the superunitary group $U(k|k)$, $\omega$ is diagonal, and $\eta$ is the diagonal matrix similar to $\beta$. Using this integral identity gives us

$$Z_2(z, \tau; z', -\tau) \sim \left(\frac{M}{2\pi}\right)^2 \int d(\omega - \mathcal{E})\frac{\Delta_s(\omega)}{\Delta_s(\mathcal{E})}(\operatorname{sdet}\omega)^{-M} e^{-M\operatorname{str}(\omega-\mathcal{E})^2/2} = \frac{\det R}{\Delta_s(\mathcal{E})}, \tag{106}$$

$$R^{\alpha\alpha'} = \frac{M}{2\pi} \int_{-\infty}^{\infty} \int_{-\infty}^{\infty} \frac{d\omega_B d\omega_F}{\omega_B + iq^{(\alpha)}0^+ - i\omega_F} \exp\Big[-Mf\left(\omega_B + iq^{(\alpha)}0^+, z^{(\alpha)}\right) \\ + Mf\left(i\omega_F, z^{(\alpha')} - (-1)^{\alpha'}\tau\right)\Big], \tag{107}$$

$$f(\omega, c) = \frac{1}{2}(\omega - c)^2 + \ln\omega, \tag{108}$$

with $(q^{(1)}, q^{(2)}) = (q, q')$.

The saddle points of the integrand in Eq. (107) are at

$$\omega_B = \omega_{B\pm}^\alpha = \frac{1}{2}\left(z^{(\alpha)} \pm i\sqrt{4 - \left(z^{(\alpha)}\right)^2}\right), \tag{109}$$

$$i\omega_F = i\omega_{F\pm}^{\alpha'} = \frac{1}{2}\left(z^{(\alpha')} - (-1)^{\alpha'}\tau \pm i\sqrt{4 - \left[z^{(\alpha')} - (-1)^{\alpha'}\tau\right]^2}\right). \tag{110}$$

Since $M$ is large, the eigenvalues of $A'$ lie within the interval $(-2, 2)$; we only need to concern ourselves with the case when $|\Re(z)| < 2$. This means that the terms under the square roots in the equations above will always have a positive real part. Note that, due to the presence of a singularity at $\omega_B = 0$, only one saddle point value is reachable through deformation of the contour of integration for $\omega_B$. This is $\omega_{B+}^\alpha$ when $q = 1$ and $\omega_{B-}^\alpha$ when $q = -1$. We need to consider both possible saddle point values for $i\omega_F$.

The result of the saddle point approximation is

$$R^{\alpha\alpha'} = \sum_\beta L_{q^{(\alpha)}\beta}^{\alpha\alpha'} \exp\left(MF_{q^{(\alpha)}\beta}^{\alpha\alpha'}\right), \tag{111}$$

$$F_{\beta\beta'}^{\alpha\alpha'} = -f\left(\omega_{B\beta}^\alpha, z^{(\alpha)}\right) + f\left(i\omega_{F\beta'}^{\alpha'}, z^{(\alpha')} - (-1)^{\alpha'}\tau\right), \tag{112}$$

$$L_{\beta\beta'}^{\alpha\alpha'} = \left(\omega_{B\beta}^\alpha - i\omega_{F\beta'}^{\alpha'}\right)^{-1} \left\{\left[1 - \left(\omega_{B\beta}^\alpha\right)^{-2}\right]\left[1 - \left(i\omega_{F\beta'}^{\alpha'}\right)^{-2}\right]\right\}^{-1/2}, \tag{113}$$

where $\beta, \beta' = \pm$. Expanding about infinite $N$ and $\tau = 0$, we find that the exponents of the exponential terms in Eq. (111) are, apart from the factor of $M$,

$$F_{\pm\pm}^{\alpha\alpha'} = \frac{(-1)^{\alpha'+1}}{2N}(x \mp 2\pi i p(x))\left\{(1 - \delta^{\alpha\alpha'})[y + i(q - q')0^+] + N\tau\right\} \\ + O(\tau^2) + O(N^{-1}\tau) + O(N^{-2}), \tag{114}$$

$$F_{\pm\mp}^{\alpha\alpha'} = \pm\left[i\pi x p(x) + \ln\left(\frac{x - 2\pi i p(x)}{x + 2\pi i p(x)}\right)\right] \\ + \frac{(-1)^{\alpha'+1}}{2N}\left\{\left[(1 - \delta^{\alpha\alpha'})x \pm \delta^{\alpha\alpha'}2\pi i p(x)\right]y + (x \pm 2\pi i p(x))N\tau\right\} \\ + O(\tau^2) + O(N^{-1}\tau) + O(N^{-2}). \tag{115}$$

The prefactors are

$$L_{\pm\pm}^{\alpha\alpha'} = \frac{(-1)^{\alpha'}N}{(1 - \delta^{\alpha\alpha'})[y + i(q - q')0^+] + N\tau} + O(\tau) + O(N^{-1}), \tag{116}$$

$$L_{\pm\mp}^{\alpha\alpha'} = \mp i(2\pi p(x))^{-2} + O(\tau) + O(N^{-1}). \tag{117}$$

Writing these quantities in this way is useful for times not larger than the Heisenberg time, where $|\tau| \ll 1$. We can restrict ourselves to these timescales since the SFF is already known to be 1 at later times.

Recall that we are interested in the $P^{\text{th}}$ power of the two-point generating function to use in Eq. 94. If $|\tau| \gg N^{-1}$, indicating times larger than the Thouless time, the $P^{\text{th}}$ power of the two-point generating function will be dominated by the saddle points that maximize the real part of $F_{\beta\beta'}^{\alpha\alpha'}$, excluding the unphysical combinations of saddle points that lead to the two-point generating function having a complex phase that grows with $N$. Considering this and using Eqs. (114)-(115), we find that the dominant Fermi-Fermi saddle point in the calculation of $R^{\alpha\alpha'}$ is $i\omega_{F\beta^*(\alpha')}^\alpha$, where

$$\beta^*(\alpha') = \delta_{qq'}q + (1 - \delta_{qq'})(-1)^{\alpha'+1}. \tag{118}$$

Our result for the $P^{\text{th}}$ power of the two-point generating function at these timescales is then

$$[Z_2(z, \tau; z', -\tau)]^P = [\Delta_s(\mathcal{E})]^{-P}\left[\det\left(L_{q^{(\alpha)}\beta^*(\alpha')}^{\alpha\alpha'}\right)\right]^P \exp\left[N\left(F_{q\beta^*(1)}^{11} + F_{q'\beta^*(2)}^{22}\right)\right], \tag{119}$$

where we have used the fact that $F_{\beta_1\beta_2}^{11} + F_{\beta_3\beta_4}^{22} = F_{\beta_1\beta_4}^{12} + F_{\beta_3\beta_2}^{21}$ to extract the exponential term from the determinant. We can verify that the normalization is correct by setting $\tau = 0$ and finding that the above expression is equal to 1, as we know it must be from Eq. (95).

If $|\tau| \gg N^{-1}$, indicating times not larger than the Thouless time, the real parts of the exponent of $R^{\alpha\alpha'}$ will not be much larger than $P^{-1}$, so they will not contribute to the determination of the dominant saddle points for the $P^{\text{th}}$ power of the two-point generating function. The saddle points that will dominate are instead determined by the prefactors $L^{\alpha\alpha'}_{\beta\beta'}$. We see from Eqs. (116)-(117) that, at these timescales, the prefactor for the $\beta = \beta'$ case is always greater than that for the $\beta = -\beta'$ case by at least a factor of $N$, so the non-dominant contributions can be neglected. This means that the dominant Fermi-Fermi saddle point in the calculation of $R^{\alpha\alpha'}$ is simply $i\omega^{\alpha'}_{Fq}$. With these considerations we find that the $P^{\text{th}}$ power of the two-point generating function at these timescales is

$$
\begin{aligned}
[Z_2(z, \tau; z', -\tau)]^P &= [\Delta_s(\mathcal{E})]^{-P} \left\{ \det\left[ L^{\alpha\alpha'}_{q^{(\alpha)}q^{(\alpha)}} \exp\left( M F^{\alpha\alpha'}_{q^{(\alpha)}q^{(\alpha)}} \right) \right] \right\}^P \\
&= e^{-i\pi N\tau(q-q')p(x)} \left[ 1 - \left( \frac{N\tau}{y + i(q-q')0^+ + N\tau} \right)^2 \right]^{-P} \\
&\quad \times \left[ 1 - \left( \frac{N\tau}{y + i(q-q')0^+ + N\tau} \right)^2 e^{i\pi(q-q')p(x)(y+2N\tau)/P} \right]^P,
\end{aligned}
$$
(120)

where we have neglected the vanishing terms in the second equality. Once again we can verify that the normalization is correct by noting that the above expression becomes 1 when $\tau = 0$.

## 5.2 The spectral form factor

We are now in a position to calculate the SFF. We begin with the case where time is not larger than the Thouless time. Inserting Eq. (120) into the correlation function of Eq. (94), then extracting the SFF using Eq. (29), we obtain

$$
K^{(\gamma)}(T) = 1 + \frac{e^{-\sigma t^2}}{2\pi i(N\tau)^2 p(x)} \sum_q \left\{ e^{-2\pi iqN\tau p(x)} \int \frac{dy}{2\pi i} e^{iN^{-1}ty} \left[ 1 - \left( \frac{N\tau}{y + N\tau + iq0^+} \right)^2 \right]^{1-P} \right.
$$
$$
\times \left[ 1 - \left( \frac{N\tau}{y + N\tau + iq0^+} \right)^2 e^{2\pi iqp(x)(y+2N\tau)/P} \right]^P - \left. \int \frac{dy}{2\pi i} e^{iN^{-1}ty} \left[ 1 - \left( \frac{N\tau}{y + N\tau} \right)^2 \right] \right\}. \quad (121)
$$

The first term in the braces is the $q = -q'$ contributions while the second term is the $q = q'$ contributions. The integral over $y$ in the second term never has singularities in the half of the complex plane in which the contour of integration may be closed, so the term vanishes.

Making the changes of variable $q \to -q$ and then $y \to qy$ and expanding with the binomial theorem, we obtain

$$
K^{(\gamma)}(T) = 1 + \frac{e^{-\sigma t^2}}{2\pi i(N\tau)^2 p(x)} \sum_q e^{2\pi iqN\tau p(x)} \int \frac{dy}{2\pi i} e^{iN^{-1}tqy} \left[ \frac{(y + qN\tau)^2}{(y - i0^+)(y + 2qN\tau)} \right]^{P-1}
$$
$$
\times \sum_{j=0}^{P} \binom{P}{j} \left( \frac{N|\tau|}{y + qN\tau} \right)^{2j} e^{-2\pi ip(x)(y+2qN\tau)j/P}. \quad (122)
$$

We observe that the integrand contains no singularities in the half of the complex plane in which the contour integral may be closed when $tq < 0$. So we can remove the sum over $q$ by

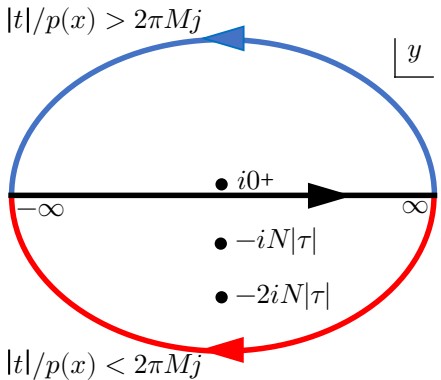

Figure 7: The poles and contours of integration for the integrals over $y$ in Eqs. (123) and (136). If $|t|/p(x)$ is less (greater) than $2\pi Mj$ the contour is closed in the lower (upper) half of the complex plane.

making the substitution $tq \to |t|$. This gives us

$$K^{(\gamma)}(T) = 1 + \frac{e^{-\sigma t^2 - 2\pi N|\tau|p(x)}}{2\pi i (N\tau)^2 p(x)} \sum_{j=0}^{P} \binom{P}{j} \int \frac{dy}{2\pi i} D_j(y), \tag{123}$$

$$D_j(y) = (N|\tau|)^{2j} \frac{(y + iN|\tau|)^{2(P-1-j)}}{(y - i0^+)^{P-1}(y + 2iN|\tau|)^{P-1}} \exp\left[iN^{-1}|t|y - 2\pi i p(x)(y + 2iN|\tau|)j/P\right]. \tag{124}$$

By examining the $y$-dependent terms in the exponent of the exponential term of Eq. (124), we see that the contour of integration can be closed in the lower half of the complex plane when $|t|/p(x) < 2\pi Mj$. In this case there may be contributions from residues located at $y = -iN|\tau|$ and $y = -2iN|\tau|$. When $|t|/p(x) > 2\pi Mj$ the contour of integration may be closed in the upper half of the complex plane and there may be a contribution from the residue at $y = i0^+$. Fig. 7 shows the contours of integration for each situation and the poles which they enclose. We find that

$$K^{(\gamma)}(T) = 1 + \frac{e^{-\sigma t^2 - 2\pi N|\tau|p(x)}}{2\pi i (N\tau)^2 p(x)} \sum_{j=0}^{P} \binom{P}{j} \begin{cases} -\operatorname*{Res}_{y=-iN|\tau|} D_j(y) - \operatorname*{Res}_{y=-2iN|\tau|} D_j(y), & |t|/p(x) < 2\pi Mj, \\ \operatorname*{Res}_{y=i0^+} D_j(y), & |t|/p(x) > 2\pi Mj. \end{cases} \tag{125}$$

We first find the residue at $y = -iN|\tau|$. Observing that it is nonzero only when $j = P$, we find that

$$\operatorname*{Res}_{y=-iN|\tau|} D_P(y) = i(N|\tau|)^2 (N^{-1}|t| - 2\pi p(x)) e^{\sigma t^2 + 2\pi N|\tau|p(x)}. \tag{126}$$

The other two residues are

$$\operatorname*{Res}_{y=-2iN|\tau|} D_j(y) = 2i(-1)^{j+1} N|\tau| e^{2\sigma t^2} \sum_{k=0}^{P-2} J(j,k) \left(\sigma t^2 - 2\pi j M|\tau|p(x)\right)^k, \tag{127}$$

$$\operatorname*{Res}_{y=i0^+} D_j(y) = 2i(-1)^{j} N|\tau| e^{4\pi Mj|\tau|p(x)} \sum_{k=0}^{P-2} J(j,k) \left(2\pi j M|\tau|p(x) - \sigma t^2\right)^k, \tag{128}$$

where

$$J(j,k) = \frac{2^{k-2(P-1)}}{k!} \sum_{k'=0}^{P-2-k} 2^{k'} \binom{2(P-1-j)}{k'} \binom{1-P}{P-2-k-k'}. \tag{129}$$

Using Eqs. (126)-(128) in Eq. (125) and unfolding by setting

$$u = |t|/p(x) = N|T|,\qquad(130)$$

$$\Lambda = \lambda p(x),\qquad(131)$$

we find that the SFF is

$$K^{(\gamma)}(T) = \min\left(\frac{T}{2\pi}, 1\right) - \frac{e^{-2\pi\Lambda^2 N^{1-\gamma}u}}{\pi\Lambda^2 N^{1-\gamma}u}\sum_{j=0}^{P}(-1)^j\binom{P}{j}Q_j^{(\gamma)}(u),\qquad(132)$$

$$Q_j^{(\gamma)}(u) = \sum_{k=0}^{\infty} J(j,k)\left(-\Lambda^2 N^{-\gamma}u|u-2\pi Mj|\right)^k \begin{cases} e^{\Lambda^2 N^{-\gamma}u^2}, & u \le 2\pi Mj, \\ e^{\Lambda^2 N^{-\gamma}u(4\pi Mj-u)}, & u \ge 2\pi Mj. \end{cases}\qquad(133)$$

Note that we have raised the upper limit for the summation index $k$ from $P-2$ to $\infty$. We can do this because $J(j,k) = 0$ for $k > P-2$. Doing so removes the need for separate treatments of the cases where $P < 3$. Although we have derived this result for times not larger than the Thouless time, it turns out to be valid at all relevant timescales, as we will show.

We now examine the SFF at times larger than the Thouless time, where $|\tau| \gg N^{-1}$. We obtain the SFF by inserting Eq. (119) into the correlation function of Eq. (94) then extracting the SFF using Eq. (29). This yields

$$K(T) = -\frac{e^{-\sigma t^2}}{2\pi i(N\tau)^2 p(x)}\sum_{qq'}qq'\int\frac{dy}{2\pi i}e^{iN^{-1}tq}\left[1-\left(\frac{N\tau}{y+N\tau+i(q-q')0^+}\right)^2\right]$$
$$\times\left[\Delta_s(\mathcal{E})\right]^{-P}\left[\det\left(L_{q^{(\alpha)}\beta^*(\alpha')}^{\alpha\alpha'}\right)\right]^P\exp\left[N\left(F_{q\beta^*(1)}^{11}+F_{q'\beta^*(2)}^{22}\right)\right].\qquad(134)$$

We note from Eq. (118) that, for the $q = q'$ terms, the dominant saddle points are the same as for the case examined above for earlier times. This means that, although there may be additional nonvanishing terms in the exponent of the exponential term of the integrand, they will be subleading and the residue analysis is unchanged. There are no singularities in the half of the complex plane in which the contour of integration may be closed. As they did for earlier times, the $q = q'$ terms of the SFF vanish.

We next consider the $q = -q' = 1$ term. From Eqs. (114) and (116)-(117) we find that $(L_{++}^{11}L_{--}^{22}-L_{+-}^{12}L_{-+}^{21})^P$ and $N(F_{++}^{11}+F_{--}^{22})$ have vanishing $y$-dependence. If we consider only positive times, which we can do since the SFF is symmetric in time, we find that the integrand once again has no singularities in the half of the complex plane in which the contour of integration may be closed. For this reason, the $q = -q' = 1$ contribution to the SFF also vanishes.

Finally we consider the only nonvanishing term, for which $q = -q' = -1$. From Eqs. (116)-(117) we find that

$$L_{-+}^{11}L_{+-}^{22} - L_{--}^{12}L_{++}^{21} = X + \left(\frac{N}{y-i0^++N\tau}\right)^2,\qquad(135)$$

where $X$ is a quantity of order 1 with its largest $y$-dependent terms being of order $N^{-1}$, so they can be neglected. Expanding with the binomial theorem, we find that the SFF is

$$K^{(\gamma)}(T) = 1 + \frac{e^{-\sigma t^2}}{2\pi i p(x)(N\tau)^2}\sum_{j=0}^{P}\binom{P}{j}\left(|\tau|^2 X\right)^{P-j}\int\frac{dy}{2\pi i}D_j'(y),\qquad(136)$$

$$D_j'(y) = (N|\tau|)^{2j}\frac{(y+iN|\tau|)^{2(P-1-j)}}{(y-i0^+)^{P-1}(y+2iN|\tau|)^{P-1}}\exp\left[iN^{-1}|t|y+N(F_{-+}^{11}+F_{+-}^{22})\right].\qquad(137)$$

From Eq. (115) we learn that that the largest $y$-dependent term in $N(F^{11}_{-+} + F^{22}_{+-})$ is $-2\pi i p(x)y$. This means that the contour of integration can be closed in the lower half of the complex plane when $|t|/p(x) < 2\pi N$. In this case there may be contributions from residues located at $y = -iN|\tau|$ and $y = -2iN|\tau|$. When $|t|/p(x) > 2\pi N$ the contour of integration is instead closed in the upper half of the complex plane and there may be a contribution from the residue at $y = i0^+$. So

$$K^{(\gamma)}(T) = 1 + \frac{e^{-\sigma t^2}}{2\pi i p(x)(N\tau)^2} \sum_{j=0}^{P} \binom{P}{j} \left(|\tau|^2 X\right)^{P-j}$$
$$\times \begin{cases} - \underset{y=-iN|\tau|}{\text{Res}} D'_j(y) - \underset{y=-2iN|\tau|}{\text{Res}} D'_j(y), & |t|/p(x) < 2\pi N, \\ \underset{y=i0^+}{\text{Res}} D'_j(y), & |t|/p(x) > 2\pi N. \end{cases} \tag{138}$$

It turns out that only the residue located at $y = -iN|\tau|$ contributes for large $N$. The exponent of the exponential factor of the terms coming from the residue at $y = -2iN|\tau|$ is

$$-\sigma t^2 + \left[iN^{-1}|t|(y - i0^+) + N(F^{11}_{-+} + F^{22}_{+-})\right]_{y=-2iN|\tau|} = \sigma t^2 - 2\pi N|\tau|p(x) + O(N\tau^2), \tag{139}$$

which becomes infinitely negative when $|t|/p(x) < 2\pi N$. Similarly, the exponent of the exponential factor of the terms coming from the residue at $y = i0^+$ is

$$-\sigma t^2 + \left[iN^{-1}|t|(y - i0^+) + N(F^{11}_{-+} + F^{22}_{+-})\right]_{y=i0^+} = -\sigma t^2 + 2\pi N|\tau|p(x) + O(N\tau^2), \tag{140}$$

which becomes infinitely negative when $|t|/p(x) > 2\pi N$. The remaining residue at $y = -iN|\tau|$ is only nonzero when $j = P$. This residue is, to leading order,

$$\underset{y=-iN|\tau|}{\text{Res}} D'_P(y) = i(N|\tau|)^2 (N^{-1}|t| - 2\pi p(x)) e^{\sigma t^2}. \tag{141}$$

This leads to the same contribution to the SFF as the residue at the same location in the case examined above for earlier times. With this result we find that the SFF for times larger than the Thouless time is

$$K^{(\gamma)}(T) = \min\left(\frac{T}{2\pi}, 1\right). \tag{142}$$

Noting that $u \gg N^{\gamma-1}$ at these timescales, we see that this is consistent with the SFF we found for earlier times in Eq. (132). Therefore Eq. (132) holds for all relevant timescales.

In Fig. 8 we compare this result for the SFF to numerical results obtained through exact diagonalization for $\gamma = 1.6$, $P = 20$ blocks, and various values of the coupling parameter $\Lambda$. When numerically calculating the SFF with the full spectrum there is good agreement between theory and numerics at early times, but the agreement is less good at later times. We can trade this early-time agreement with later-time agreement by using only the eigenvalues within a window smaller than the spectral width. We can combine these results to get good agreement at all times of interest by determining the time at which the two approaches converge, then using the full spectrum result before that time and the windowed results at later times. We observe that the SFF decays from the fully uncoupled result to the GUE SFF over time, with the rate of decay controlled by $\Lambda$. As we vary $\Lambda$ we can see the SFF moving between the behaviors we predicted schematically in Fig. 2. As in the RP model, for small values of $\Lambda$ we see that the SFF reaches its plateau at times greater (but not much greater) than the Heisenberg time. The agreement between theory and numerics is still very good at these late times.

Somewhat similar SFFs with a short-time peak then a crossover to RMT behavior have been found for random quantum circuits [66–68] and the mass-deformed Sachdev-Ye-Kitaev model [69]. However, the enhanced ramps are faster than linear in time in these cases because these models transition from many-body-localized to ergodic behavior, in contrast to the glassy to ergodic transition we see in the BRP model.

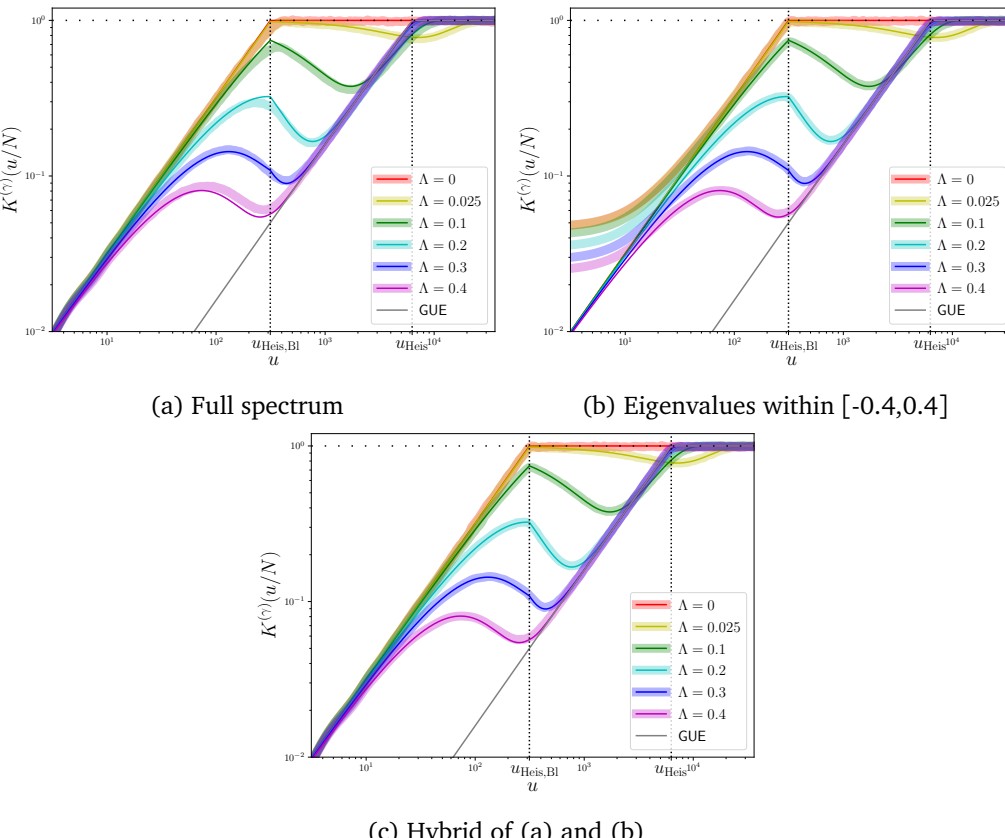

(a) Full spectrum

(b) Eigenvalues within [-0.4,0.4]

(c) Hybrid of (a) and (b)

Figure 8: The SFF of the BRP model when $\gamma = 1.6$ and there are $P = 20$ blocks for several values of the coupling parameter $\Lambda$. The dark thin lines show the analytical results while the lighter, thicker lines show numerical results. The left (right) dotted line indicates the block (full) Heisenberg time. The numerics are obtained through exact diagonalization of size $N = 1000$ random matrices and are averaged over $10^5$ realizations. In (a) the numerics are calculated with the full spectrum while in (b) only the eigenvalues within the window [-0.4.0.4] are used. In (c) the results of (a) are used at times before the numerical curves for a particular $\Lambda$ converge and the numerical results of (b) are used after.

## 5.3 The infinite matrix size limit

We now seek to find the behavior of the SFF in the large $N$ limit for all timescales of interest. Above we have already considered the SFF at times greater than the Thouless time and argued that several terms vanish at large $N$, so Eq. (142) is the large $N$ limit for the SFF at those timescales. Going forward we will consider earlier times. We will assume for now that we are not at the Thouless and Heisenberg timescales simultaneously, which occurs when $\gamma = 2$ and $u \sim t_{\text{Th}} \sim N$. This is a special case which we will examine later.

Under these conditions, the sum of the $k = 0$ contributions to the SFF in the time period $2\pi M j \leq u \leq 2\pi M (j+1)$ is

$$\frac{e^{-2\pi\Lambda^2 N^{1-\gamma}u}}{\pi\Lambda^2 N^{1-\gamma}u}\left[\sum_{l=0}^{j}(-1)^l\binom{P}{l}J(l,0)e^{\Lambda^2 N^{-\gamma}u(4\pi M l-u)} + e^{\Lambda^2 N^{-\gamma}u^2}\sum_{l=j+1}^{P}(-1)^l\binom{P}{l}J(l,0)\right]. \quad (143)$$

Since $\gamma > 1$, the exponents of the exponential terms within the brackets are small. Expanding

them to first order, we can write this as

$$\frac{e^{-2\pi\Lambda^2 N^{1-\gamma}u}}{\pi\Lambda^2 N^{1-\gamma}u}\sum_{l=0}^{P}(-1)^l\binom{P}{l}J(l,0)$$

$$+\frac{e^{-2\pi\Lambda^2 N^{1-\gamma}u}}{\pi}\left[\sum_{l=0}^{j}(-1)^l\binom{P}{l}J(l,0)(T-4\pi l/P)-\sum_{l=j+1}^{P}(-1)^l\binom{P}{l}J(l,0)T\right]. \quad (144)$$

We note that $J(j,k)$ is a polynomial of order less than $P$ in $j$. So, from the theory of finite differences, we find that [70]

$$\sum_{j=0}^{P}(-1)^j\binom{P}{j}J(j,k)=0. \quad (145)$$

Using this fact, we find that the sum of the $k < 2$ contributions is, to first order in $N^{-1}$,

$$\frac{2}{\pi}e^{-2\pi\Lambda^2 N^{1-\gamma}u}\sum_{l=0}^{j}(-1)^l\binom{P}{l}[J(l,0)+J(l,1)](T-2\pi l/P). \quad (146)$$

The sum of the $k \geq 2$ contributions will be subleading. So, to first order, the SFF in this time range is

$$K^{(\gamma)}(T)=\min\left(\frac{T}{2\pi},1\right)+\frac{2}{\pi}e^{-2\pi\Lambda^2 N^{1-\gamma}u}\sum_{l=0}^{j}(-1)^l\binom{P}{l}[J(l,0)+J(l,1)](T-2\pi l/P). \quad (147)$$

We now seek to simplify $J(l,0)+J(l,1)$. By shifting the summation index in $J(l,1)$ and using Pascal's rule

$$\binom{n-1}{k}+\binom{n-1}{k-1}=\binom{n}{k}, \quad (148)$$

we find that

$$J(l,0)+J(l,1)=2^{-2(P-2)}\sum_{k'=0}^{P-2}2^{k'}\binom{2(P-1-l)+1}{k'}\binom{1-P}{P-2-k'}=J(l-1/2,0). \quad (149)$$

We can also find a closed-form expression for $J(l,0)$ by noting that it is the $(P-2)^{\text{th}}$ coefficient in the Maclaurin series expansion

$$J(l,0)=2^{-2(P-1)}[z^{P-2}](1+z)^{1-P}\sum_{k=0}^{\infty}(2z)^k\binom{2(P-1-l)}{k}, \quad (150)$$

where $[z^{P-2}]$ is the $(P-2)^{\text{th}}$ coefficient extractor. Now, summing over $k$, writing the coefficient as a residue at $z = 0$, and making the change of variable $z \to (\sqrt{4z+1}-1)/2$, we find that

$$J(l,0)=2^{-2(P-1)}\operatorname*{Res}_{z=0}z^{1-P}(1+4z)^{P-l-3/2}=\frac{1}{4}\binom{P-l-3/2}{P-2}. \quad (151)$$

Using Eqs. (149) and (151) in Eq. (147), we find, for $2\pi j/P \leq T \leq 2\pi(j+1)/P$,

$$K^{(\gamma)}(T)=\min\left(\frac{T}{2\pi},1\right)+e^{-2\pi\Lambda^2 N^{1-\gamma}u}\begin{cases}(P-1)\frac{T}{2\pi}, & j=0,\\ 1-\frac{T}{2\pi}, & 0<j<P,\\ 0, & j=P.\end{cases} \quad (152)$$

This means that, overall, the SFF is

$$K^{(\gamma)}(T) = \left(1 - e^{-2\pi\Lambda^2 N^{1-\gamma} u}\right) \min\left(\frac{T}{2\pi}, 1\right) + e^{-2\pi\Lambda^2 N^{1-\gamma} u} \min\left(P\frac{T}{2\pi}, 1\right). \tag{153}$$

This result becomes 1 for times greater than the Heisenberg time, so it is also valid at those times. We conclude that Eq. (153) is valid at all relevant times, excepting the case of coincident Thouless and Heisenberg timescales. For times smaller than the Thouless time, where $u \ll N^{\gamma-1}$, we see that

$$K^{(\gamma)}(T) = \min\left(P\frac{T}{2\pi}, 1\right). \tag{154}$$

This result is exactly what we would expect in the weak-coupling limit: the GUE SFF with time enhanced by a factor of $P$, the number of blocks.

For the Thouless timescale, while different from the Heisenberg timescale (meaning $\gamma \neq 2$), we obtain

$$K_{\text{Th}}^{(\gamma)}(v) = K^{(\gamma)}(N^{\gamma-2}v) = \left(1 - e^{-2\pi\Lambda^2 v}\right) \min\left(\frac{T}{2\pi}, 1\right) + e^{-2\pi\Lambda^2 v} \min\left(P\frac{T}{2\pi}, 1\right), \tag{155}$$

where $v = N^{2-\gamma}T$. We see that at the Thouless timescale there is a crossover from the GUE SFF for large $\Lambda$ to the uncoupled blocks result for small $\Lambda$. For times greater than the block Heisenberg time $2\pi M$, this becomes identical to the RP SFF. The reason for this is that the correlations between the eigenvalues of $A$ do not persist for times larger than the block Heisenberg time. At these timescales we are justified in setting $Z_2(z, \tau; z', -\tau) = [g_2(z, \tau; z', -\tau)]^M$ in Eq. (94), which reduces the calculation of the SFF to that of the RP model.

We now examine the $\gamma = 2$ case at the Thouless timescale. From Eq. (132) we find that

$$K_{\text{Th}}^{(2)}(v) = K^{(2)}(v) = \min\left(\frac{v}{2\pi}, 1\right) - \frac{e^{-2\pi\Lambda^2 v}}{\pi\Lambda^2 v} \sum_{j=0}^{P} (-1)^j \binom{P}{j} Q_{\text{Th},j}^{(2)}(v), \tag{156}$$

$$Q_{\text{Th},j}^{(2)}(v) = \sum_{k=0}^{P-2} J(j,k)(-\Lambda^2 v |v - 2\pi j/P|)^k \begin{cases} e^{\Lambda^2 v^2}, & v \leq 2\pi j/P, \\ e^{\Lambda^2 v(4\pi j/P - v)}, & v \geq 2\pi j/P. \end{cases} \tag{157}$$

Due to the coincidence of the Thouless and Heisenberg timescales, the SFF in this case does not generally have a closed form expression. Although the SFF is unwieldy in general, it does simplify greatly for $P < 3$. When $P = 1$ we find the GUE result

$$K_{\text{Th}}^{(2)}(v) = \min\left(\frac{v}{2\pi}, 1\right), \tag{158}$$

while for $P = 2$, the simplest nontrivial case, we find

$$K_{\text{Th}}^{(2)}(v) = \min\left(\frac{v}{2\pi}, 1\right) + \frac{e^{-2\pi\Lambda^2 v}}{2\pi\Lambda^2 v} \begin{cases} \sinh(\Lambda^2 v^2), & 0 \leq v \leq \pi, \\ e^{\Lambda^2 v(2\pi-v)} - \cosh(\Lambda^2 v^2), & \pi \leq v \leq 2\pi, \\ -2\sinh^2(\pi\Lambda^2 v)e^{\Lambda^2 v(2\pi-v)}, & v \geq 2\pi. \end{cases} \tag{159}$$

The Thouless-Heisenberg timescale SFF is plotted in Fig. 9, for the cases of $P = 4$ and $P = 20$. Also shown are numerical results obtained through exact diagonalization which are in strong agreement with the theory. For times much greater than the block Heisenberg time there is graphical evidence that the SFF moves to that of the RP model. By examining Fig. 9, we can see that when the number of sectors $P$ becomes large, making $u \gg 2\pi M$, the SFF moves to the RP SFF shown in Fig. 5. As with the RP SFF, we see that the SFF reaches its plateau value at times greater than the Heisenberg time when $\Lambda$ is finite. In this case there is

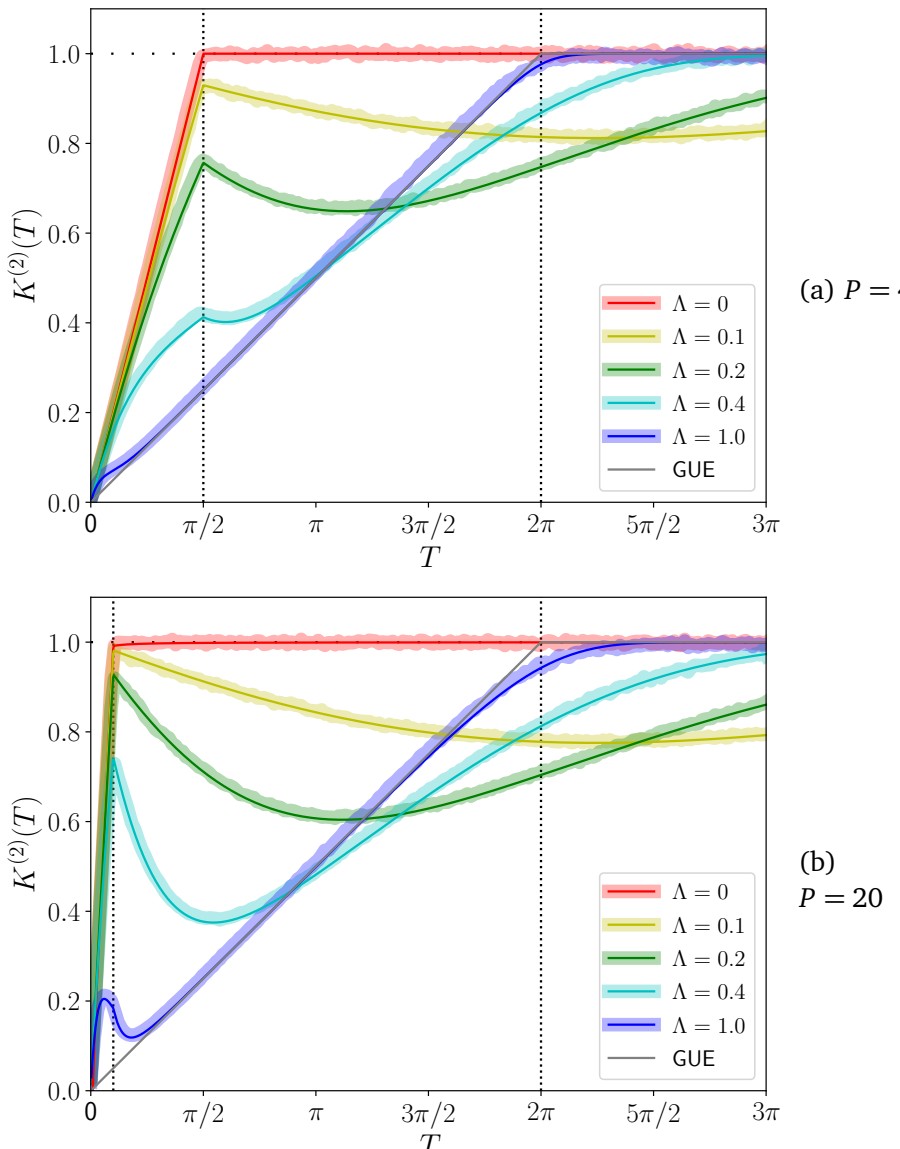

Figure 9: The SFF for the BRP model when the Thouless and Heisenberg timescales coincide for various values of the coupling parameter Λ. The dark thin lines show the analytical results while the lighter, thicker lines show the numerical results. The numerics are obtained through exact diagonalization of size $N = 1000$ random matrices and are averaged over $10^5$ realizations. The numerics are restricted to eigenvalues within the window [-0.75,0.75]. In (a) there are $P = 4$ blocks while in (b) there are $P = 20$. The left (right) dotted line indicates the block (full) Heisenberg time.

also a trade-off between level repulsion at early and late times. The area between the SFF and its plateau value is independent of the coupling strength for any nonzero coupling between the sectors, as is shown in the Appendix.

Bringing together all the results at the Thouless timescale we have that the SFF is

$$K_{\text{Th}}^{(\gamma)}(\nu) = \begin{cases} \min\left(P\frac{T}{2\pi}, 1\right), & \gamma > 2, \\ \min\left(\frac{\nu}{2\pi}, 1\right) - \frac{e^{-2\pi\Lambda^2\nu}}{\pi\Lambda^2\nu}\sum_{j=0}^{P}(-1)^j\binom{P}{j}Q_{\text{Th},j}^{(2)}(\nu), & \gamma = 2, \\ \left(1 - e^{-2\pi\Lambda^2\nu}\right)\min\left(\frac{T}{2\pi}, 1\right) + e^{-2\pi\Lambda^2\nu}\min\left(P\frac{T}{2\pi}, 1\right), & 1 < \gamma < 2. \end{cases} \tag{160}$$

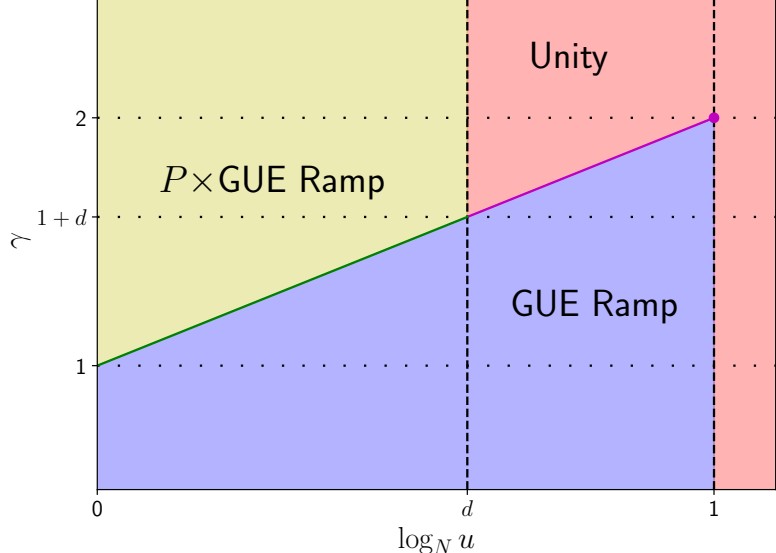

Figure 10: The SFF of the BRP model at different timescales and values of the coupling parameter $\gamma$ when the size of the blocks $M$ is much less than $N$. The green line represents a crossover between the enhanced GUE ramp and the regular GUE ramp. The magenta line represents the crossover between the Poissonian result and the regular GUE ramp. The magenta dot indicates the point at which the Thouless and Heisenberg timescales coincide and there is no closed form expression for the SFF. The dotted lines indicate the values of $\gamma$ at which there is a phase transition. The left (right) dashed line indicates the block (full) Heisenberg time.

The behavior of the SFF is dependent of $\Lambda$ when $1 < \gamma \leq 2$. For small $\Lambda$ the SFF goes to the result expected for uncoupled sectors, while for large $\Lambda$ it goes to the GUE result. This can be seen explicitly in the above equation for $1 < \gamma < 2$ and can be observed graphically for the $\gamma = 2$ case in Fig. 9.

As we did for the RP model, we can create a diagram showing the behavior of the SFF at different timescales and values of $\gamma$, shown in Fig. 10. At times larger than the Thouless time the SFF goes to the GUE result. At times smaller than the Thouless time the SFF goes to the GUE SFF with time enhanced by a factor of $P$, the number of blocks. If $\gamma > 1 + d$ this result first reaches its plateau value at the block Heisenberg time, which is smaller than the full Heisenberg time by a factor of $P$. This means that at times larger than the block Heisenberg time the SFF is the same as the RP SFF. At the Thouless timescale, while it is not larger than the Heisenberg timescale, there is a crossover between the enhanced GUE SFF and the standard GUE SFF. The SFF is 1 at times larger than the Heisenberg time.

# 6 Conclusion

In this paper we introduce a generalization of the Rosenzweig-Porter (RP) model called the Block Rosenzweig-Porter (BRP) model. This is done by redefining the diagonal matrix in the RP model to be a block diagonal matrix with each block being a GUE matrix. In doing so we obtain a minimal quantum glass model which immediately thermalizes within the blocks but has much slower global thermalization (if it thermalizes at all) depending on the strength of the inter-block coupling. It is known that the RP model is chaotic for $\gamma < 1$ and localized for $\gamma > 2$. For intermediate values of $\gamma$ the RP model is localized at early times, then thermalizes

and becomes chaotic at the Thouless time. We find that the same is true for the BRP model. However, the localized behaviors for the two models are different. Whereas the RP model exhibits localization in single states, the BRP model instead exhibits localization within single blocks.

Our main result is a calculation of the spectral form factor (SFF) at all timescales larger than the inverse spectral width for $\gamma > 1$ in the BRP model. As a lead-in to this, we perform this calculation for the RP model. In the intermediate phase where $1 < \gamma < 2$, the SFF of the RP model indicates Poissonian statistics before the Thouless timescale and GUE statistics afterward. At the Thouless timescale there is a crossover between these behaviors mediated by the unfolded coupling parameter $\Lambda$. These results are consistent with those of Ref. [27]. Within the same range for $\gamma$, the BRP model has statistics consistent with independent GUE blocks before the Thouless timescale. After the Thouless timescale statistics consistent with a single large GUE block are found. As with the RP model, there is a crossover between these behaviors at the Thouless timescale, again mediated by $\Lambda$. This indicates that the system is initially frozen into a single sector, then escapes and thermalizes at the Thouless time.

An important feature of the SFF is the times at which it is equal to its plateau value, as these indicate the disappearance of repulsion between eigenvalues at the energy separations corresponding to these times. We find that if $\gamma < 1 + d$, where $d = \log_N M$, the SFF will first reach its plateau value at or near the Heisenberg time, meaning that eigenvalue correlations first vanish at separations smaller than the mean level spacing, as they must for all discrete quantum systems. However, if $\gamma > 1 + d$ the SFF first reaches its plateau value at the block Heisenberg time, which is smaller than the full Heisenberg time by a factor of $P$, the number of blocks. This means that correlations between eigenvalues first vanish at separations smaller than the mean level spacing of a single block. If $\gamma < 2$, meaning the system is not in its localized phase, level repulsion may still be present at later times (smaller energy separations), causing the SFF to drop back below its plateau value until the Heisenberg time.

While this work has been concerned with dynamics and associated spectral statistics, one can also ask about the eigenstate properties in the three phases. In the RP model the eigenstates are fully ergodic for $\gamma < 1$ and localized to a single state for $\gamma > 2$. For intermediate values of $\gamma$ the eigenstates are neither localized nor ergodic; they are termed nonergodic extended states [27]. A similar phenomenon occurs in the BRP model. For $\gamma > 2$ the eigenstates are localized to a single sector. However, the eigenstates become fully delocalized across the Hilbert space for $\gamma < 1 + d$ (rather than merely $\gamma < 1$), where $M$ is the sector size. The fact that this eigenstate transition is, in general, separate from the transition to full GUE statistics at $\gamma = 1$ is an interesting feature of the BRP model. Future work may examine this feature and the eigenstate statistics in more detail.

The BRP model we introduce here is significant because it is a solvable random matrix model with a glass transition. The SFF of the BRP model can be calculated at all relevant timescales. In more complex quantum glass models one cannot probe times exponential in the system size with currently available methods. Particularly interesting is the case in which the Thouless time is of the same order as the Heisenberg time. The BRP model is thus a useful starting point in understanding the spectral statistics of systems with slow dynamics.

Due to the simplicity of the BRP model it does not capture some features of more physical models. Each block is taken to be the same size and independent of the others. More generally we can expect the sectors to have some distribution in size with correlations between their matrix elements. The BRP Hamiltonian also has only two levels in its hierarchy, making it a model with a single nearly conserved quantity or slow mode. More realistic systems may have a richer hierarchy of long timescales, which could be captured by a generalized BRP model with more than two levels of nested blocks. An avenue for future work may be to modify the BRP model to capture these features.

# Acknowledgements

We would like to thank I. M. Khaymovich for useful discussion and comments which improved this manuscript, in particular for pointing out the interpretation of the coupling parameter $\gamma$ in terms of eigenstate properties.

**Funding information** This work was supported by the following: The National Science Foundation through the Quantum Leap Challenge Institute for Robust Quantum Simulation (grant OMA-2120757), NSF DMR-2037158, US-ARO Contract No.W911NF1310172, and Simons Foundation (V.G.); the Joint Quantum Institute (M.W.); the Air Force Office of Scientific Research under award numbers FA9550-17-1-0180 (M.W.) and FA9550-19-1-0360 (B.S.); the U.S. Department of Energy, Office of Science, Office of Advanced Scientific Computing Research, Accelerated Research for Quantum Computing program "FAR-QC" (R.B.); the DoE ASCR Quantum Testbed Pathfinder program under award number DE-SC0019040 (C.L.B.); the DoE ASCR Accelerated Research in Quantum Computing program under award number DE-SC0020312 (C.L.B.); the DoE QSA, AFOSR, AFOSR MURI, NSF PFCQC program, NSF QLCI under award number OMA-2120757 (C.L.B.); DoE award number DE-SC0019449 (C.L.B.), ARO MURI, and DARPA SAVaNT ADVENT (C.L.B.). This material is based upon work supported by the National Science Foundation Graduate Research Fellowship Program under Grant No. DGE 1840340 (R.B.), and by the National Science Foundation NRC postdoctoral fellowship program (C.L.B.).

# A  Appendix

Here we show that the area between the SFF and its plateau value is independent of the strength of the coupling between different states, as long as the coupling is nonzero. Consider the integral

$$I = \int_{-\infty}^{\infty} dt \, (\mathrm{SFF}(t) - N), \tag{A.1}$$

where

$$\mathrm{SFF}(t) = \sum_{mn} \overline{\exp[i(E_n - E_m)t]} \tag{A.2}$$

is the SFF. We can approximate $I$ as

$$I(t_{\mathrm{long}}) = \int_{-\infty}^{\infty} dt \, (\mathrm{SFF}(t) - N) \exp\left(-\frac{t^2}{2t_{\mathrm{long}}^2}\right), \tag{A.3}$$

which goes to $I$ as $t_{\mathrm{long}} \to \infty$. We find that

$$
\begin{aligned}
I(t_{\mathrm{long}}) &= \int_{-\infty}^{\infty} dt \sum_{m \neq n} \overline{\exp[i(E_n - E_m)t]} \exp\left(-\frac{t^2}{2t_{\mathrm{long}}^2}\right) \\
&= \sqrt{2\pi} t_{\mathrm{long}} \sum_{m \neq n} \overline{\exp\left(\frac{-(E_n - E_m)^2 t_{\mathrm{long}}^2}{2}\right)}.
\end{aligned}
\tag{A.4}
$$

If there is repulsion between all levels so that each level has a nonzero distance from the others, $I(t_{\mathrm{long}})$ will vanish for $t_{\mathrm{long}}$ larger than the Heisenberg time. This indicates that $I = 0$.

Because the disconnected part of the SFF depends only on the level density, which for the models considered in this work is independent of the coupling strength when $\gamma > 1$, the area

between the connected SFF and its plateau value will also be independent of the coupling strength for those values of $\gamma$. We found above that, as $\gamma \to 1^+$ for both models, the SFF goes to the GUE result, which is also the result for all $\gamma \leq 1$. From this we can conclude that the area between the unfolded connected SFF and its plateau value is equal to the GUE result for any nonzero coupling strength. This is simply a triangle with area $\pi N$.

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
