# Peer review of "Spectral statistics of a minimal quantum glass model"

_SciPost Physics, doi:SciPost Phys. 15, 084 (2023)_

## Round 2 · Referee Report · Anonymous (Referee 1) · 2023-5-17

Strengths

  • Comprehensive analysis of the spectral form factor in the RP matrix model and a new "glassy" generalization involving block matrices.
  • Good introduction and summary.
  • Nice plots show impressive agreement of numerics and analytics across many timescales.

Weaknesses

  • The model's analytical treatment is nice but not easy to follow.

Report

This paper introduces a new matrix model, which generalises the unitary Rosenzweig-Porter model to a model of a block-diagonal matrix coupled to random GUE matrices. This new "block-RP" model is analytically tractable even at exponentially large times and it exhibits a rich phase structure, going from a chaotic phase into a glassy non-ergodic phase with an interesting transition region where the system is initially localized (within blocks) and later thermalises (globally).

The analysis is very detailed and comprehensive. The spectral form factor is computed and fully analysed both numerically and analytically for the RP model first. The authors then do the same for the block-RP model. The analysis of different relevant time scales and phases is done carefully and comprehensively. Extensive calculations and results are presented in the main text, which makes the bulk of the paper somewhat tough to digest. Nevertheless, a detailed summary and introduction with schematic plots make the results accessible.

It seems that the paper was written with great care. The results are interesting as they provide a potentially useful model for studying the chaos/glass transition in a tractable model that shares some basic features with realistic systems. I recommend publication of the manuscript as it is.

---

## Round 2 · Referee Report · Anonymous (Referee 2) · 2023-5-18

Strengths

1 - Introduction of a solvable random matrix model - the block Rosenzweig-Porter model.

2 - Exact results for spectral statistics, i.e., the spectral form factor.

4 - Clear exposition of technical computations.

5 - Solid back up of analytical computations with numerical results.

Weaknesses

1 - No direct comparison with more realistic/physical models for quantum glasses.

2 - Despite the clear exposition of the derivation, some technical steps are still not easy to follow.

Report

The authors introduce a solvable random matrix model, the block Rosenzweig-Porter model, which mimics individual sectors in Hilbert space subject to a tunable interaction and models, e.g., quantum glasses. The spectral statistics of the block RP model exhibits a transition from a chaotic to a glassy regime. This is demonstrated by means of the spectral form factor for which the authors derive exact results at all time scales larger than the inverse spectral width.

Despite its technicality, the authors provide a clear exposition of their derivation and back up their results with solid numerical data. Moreover, they provide some intuition about their results, e.g., in terms of localization properties of eigenstates, thereby making the paper more accessible.

The block RP model and its spectral statistics shed some light in how random matrix behavior emerges from and depends on the interaction of individual blocks, which is both interesting on its own and potentially applies also to more realistic physical systems. I am therefore convinced that the paper easily meets the criteria for publication and recommend publication after some very minor revision.

Requested changes

1 - Below Eq. (57) the authors restrict to times smaller than the Heisenberg time for the computation of the spectral form factor (SFF) in the RP model, stating that for larger times the SFF has reached its plateau. In Fig. 4. the latter is not the case for small coupling parameter Lambda, but the expression (66) accurately describes the SFF even for times larger than Heisenberg time. Could the authors comment on why their results matches numerics so well in a time regime which seems to be excluded in the derivation of (66)?

---

## Round 3 · Referee Report · Anonymous (Referee 2) · 2023-6-29

Report

The Authors have adressed my remarks thoroughly by adding a detailed discussion to the manuscript.

I recommend publication of the paper in its current form.

---

## Round 3 · Author Response

We are grateful to the referees for their comments on our manuscript. At one referee's request we have included some discussion regarding the good agreement between the theory and numerics for the spectral form factor of the Rosenzweig-Porter model at times larger than the Heisenberg time. There is some broad discussion of differences between the Block Rosenzweig-Porter model and more physical models at the end of the final section. With the changes made we believe the manuscript is ready for publication.

---

## Round 3 · List of Changes

1) Concerning the SFF of the RP model, this sentence has been added at the end of the first paragraph on page 16: "Although it was derived considering only times not larger than the Heisenberg time, it continues to hold for later times, as we will discuss in the following section." 2) At the end of the second paragraph on page 16, which is a discussion of Fig. 4, we have added "For small values of $\Lambda$ we see that the SFF reaches its plateau at times greater (but not much greater) than the Heisenberg time. The agreement between theory and numerics is still very good at these late times." 3) The second-to-last paragraph of Section 4, including Eq. (82), has been added to show that the plateau behavior is recovered when time is taken to be larger than the Heisenberg time in our result for the RP SFF. 4) Additional minor edits.

---

## Editorial Decision

published